# Bridging Between Stable Rank and Data Selection: A Novel Sampling Method for Fast Training of Deep Neural Networks

## Abstract

Data selection for efficient training aims to reduce the computational cost by selecting a subset of data to approximate the objective function. A number of elegant approaches have been proposed in past years, such as the popular importance sampling and coreset methods. However, their required sample sizes usually have a linear dependence on the dimension of parameter space (or other types of dimensions like pseudo dimension), which could be very large and thus hinder their applications to deep neural networks. In this paper, we aim to provide a deeper understanding of the connection between data selection and the complexity of training space in theory. Inspired by the effectiveness of prevalent low-rank fine-tuning techniques, we propose to study the sample size from the perspective of Gradient Trajectory (GT). Specifically, we measure the dimension of training space by the "stable rank" of gradient trajectory matrix (GT matrix), and propose a novel data selection method called "Stable Rank related Stratified Sampling method (SRS-Sampling)" to accelerate the training process. Moreover, we establish the theoretical framework between the evolving stable rank of GT matrix and the required sample size. Finally, we conduct a set of experiments across pre-training and fine-tuning to validate the effectiveness of SRS-Sampling.

## 1 Introduction

**Data selection** for efficient training of neural networks is a critical challenge, particularly for large-scale datasets where computational costs can be prohibitive (Mirzasoleiman et al., 2020). This problem has also attracted active research interest from industry, such as the recent studies by Microsoft (Li et al., 2024) and Facebook (Touvron et al., 2021). In general, the goal of data-efficient training is to construct a small weighted proxy $Q$ for the original dataset $P$, such that the objective function evaluated on $Q$ approximates its value on $P$. Meanwhile, the computational cost of neural network training can be largely reduced since $|Q| \ll |P|$.

Coreset construction is a representative data selection method, where a portion of them select diverse data via greedy selection (Mirzasoleiman et al., 2020; Killamsetty et al., 2021a;c; Pooladzandi et al., 2022; Yang et al., 2023c). They typically compute a score based on certain criteria (*e.g.,* forgetting events (Toneva et al., 2019b), data coverage (Zheng et al., 2023) and Shapley value (Wang et al., 2024; 2025)) to quantify the importance of each sample. The scores are then used to adjust the sampling probability accordingly (Mindermann et al., 2022; Xia et al., 2022; 2024). Besides subset selection, data distillation directly synthesizes a small dataset, where the objective may involve feature matching, gradient matching, or trajectory matching (Wang et al., 2022a; Zhao et al., 2021; Cazenavette et al., 2022). Although these methods can provide an informative subset $Q$, there is usually a non-negligible computational overhead associated with the scoring or distillation process.

Importance sampling (Johnson & Guestrin, 2018) is another widely studied approach for data selection (it is also often regarded as a coreset method). Many importance sampling based methods rely on the notion of "sensitivity" (Langberg & Schulman, 2010), which measures the maximum contribution of a single data point to the overall objective function. Data points are then sampled with probability proportional to their sensitivity, and the required sample size depends on the total sensitivity, which is determined by the so-called "pseudo-dimension" (a dimension closely related to VC

dimension) of the objective function (Li et al., 2001; Feldman & Langberg, 2011). More generally, approximating the objective function within a bounded error requires a sample size that reflects the "complexity" of the model (or the objective function). Although being effective for classical machine learning tasks such as clustering (Feldman & Langberg, 2011), logistic regression (Munteanu et al., 2018), and SVMs (Tukan et al., 2020), this approach faces a fundamental challenge in deep learning: it has been proved that the pseudo dimension of a deep neural network is **at least linear in the number of parameters**, which can range from several millions to billions (Yang et al., 2023b). As a consequence, the resulting subset size required in theory will become extremely large and arguably unaffordable in practice.

In summary, it is urgently needed to consider some new perspective on model complexity for developing an efficient data selection method for deep neural networks. Our main idea in this paper (which is latter briefly introduced in Section 1.1) is partly inspired by previous works from both the theoretical and practical aspects. The recent work by Huang et al. (2021) characterizes complexity by the dimensionality of the parameter training subspace, namely the subspace spanned by the sequence of parameter updates. They showed that the required sample size scales with the dimensionality of the parameter update subspace. This suggests that *if the updates lie within a lower-dimensional subspace, fewer samples should be sufficient for model training.* Beyond this theoretical perspective, recent empirical studies provide strong evidence that effective training can often be confined to a much lower $r$-dimensional subspace with $r \ll d$, where $d$ is the dimension of the parameter space. For example, Li et al. (2018) demonstrate that training an MLP on MNIST with setting $r = 0.4\% \times d$ does not incur significant accuracy loss. Hu et al. (2022) shows that LoRA fine-tuning of GPT-3 achieves even better validation accuracy with $r = d/10000$. Zhao et al. (2024) pre-trains LLaMA using $r = 35\% \times d$ and attain comparable performance. These results highlight a striking gap between the empirical effectiveness of low-rank training and the limitations of existing theoretical analyses, reinforcing the need for a systematical study on the question that how the dimension of the training subspace governs the required sample size for data selection.

## 1.1 THE CHALLENGES AND OUR CONTRIBUTIONS

Existing low-rank fine-tuning methods restrict updates to lie in or near a low-dimensional subspace, typically via LoRA adapters or SVD decompositions of weight or gradient matrices. To develop a sample size analysis that applies broadly to general training scenarios, the key challenge is to quantify the dimensionality of the subspace in which sequential training steps evolve. While the specific updates may vary across optimizers, a simple observation is that the parameter at each timestamp $t$ is a linear combination of the initial parameter and the past gradients. Please see Fig. 1 for an illustration. As a result, the subspace spanned by the parameter updates (*i.e.,* $\theta_t - \theta_0$) is contained within the subspace spanned by the gradients. Therefore, the subspace dimension of the parameter trajectory is upper-bounded by that of the gradient trajectory. This allows us to transfer the analysis from **parameters** to **gradients**, and the observation above motivates the following question:

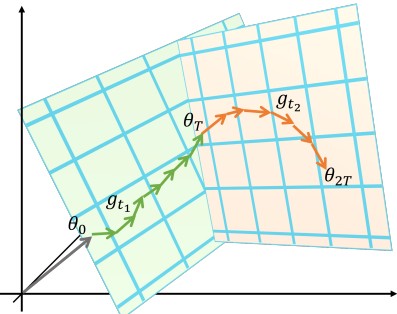

Figure 1: GT mainly resides in a low-dimensional principal subspace (the green and orange subspaces), whose dimension is characterized by the stable rank (Algorithm 2) and determines the sample size in Algorithm 1, which is updated every $T$ iterations (*i.e.,* at parameter $\theta_0, \theta_T, \theta_{2T}, \cdots$).

*How to effectively analyze the dimensionality of the subspace in which the Gradient Trajectory (GT) resides, and how can such an analysis guide the design for our data selection?*

Directly analyzing the dimension of the GT subspace is not an ideal choice. Even within a bounded local region of the parameter space, the dimension of the GT subspace can still be large, which may not be helpful to our design of sample size. Instead, we propose to study the *stable rank* (Rudelson & Vershynin, 2007) of gradient trajectory matrix (*GT matrix*), which is a continuous and soft version of rank and we defer the formal definition to Definition 1. It measures the rank of principal components in GT matrix, which corresponds to the dimension of principal space where GT mainly resides.

However, we still face the following two challenges. **First**, how to analyze the stable rank of the GT matrix for neural networks. **Second**, how to design the sample size based on the stable rank, instead of relying on the parameter dimension.

For the first challenge, we derive an upper-bound on the stable rank of GT matrix based on a consistent gradient structure across a general family of nonlinear networks (*e.g.,* ResNet with ReLU) (Tian, 2020; Zhao et al., 2024). We also generalize the analysis to Transformer architecture (Vaswani et al., 2017), under a simple intuition about the stability of attention matrix. The upper bound involves determining the minimal eigenspace where gradients converge slowest, defined by some smallest eigen-pairs of convergence operator $S$ (available as a by-product of training and introduced later).

For the second challenge, we focus on the trajectory subspace of the local training process. Although its dimension may reach $d$, it can be decomposed into two parts: the principal subspace, spanned by the singular vectors corresponding to the largest singular values of the GT matrix, captures the dominant directions of the trajectory. In contrast, the residual subspace, formed by the remaining singular vectors, captures minor variations. We establish that the dimension of the principal subspace is effectively characterized by the stable rank of the GT matrix. Based on this, we allocate a sample size proportional to the stable rank, such that a bounded approximation error is provided for a set of "anchors" in the principal subspace, meanwhile the approximation errors in the residual subspace are controlled by smaller singular values. Consequently, the required sample size scales linearly with the stable rank rather than with the full parameter dimension.

Overall, for data-efficient training of deep neural networks, we propose to obtain the weighted subset by performing **S**table **R**ank related **S**tratified **S**ampling (**SRS-Sampling**) on the original dataset. We establish a theoretical framework that linearly relates the required weighted subset size for objective function approximation to the stable rank of GT matrix, which is much smaller than the parameter dimension as validated in our experiments. The removal of dependence on $d$ in the sample size enables the scaling of SRS-Sampling to deep neural networks. One can also adaptively adjust the computational budget across training stages according to stable rank, such that the overall training process can be more efficient. Finally, we conduct a set of experiments to investigate the effectiveness of SRS-Sampling upon several popular benchmark datasets.

## 2 PRELIMINARIES

Before we introduce the high-level idea of our stable rank related stratified sampling procedure, we need to introduce some basic notation first. We assume that the objective function $F(\theta) = \frac{1}{n} \sum_{i=1}^{n} f(\theta, x_i, y_i)$ is Lipschitz smooth, which is a common assumption for analyzing many gradient descent methods (Wolfe, 1969).

**Assumption 1** (Lipschitz Smoothness)**.** *There exists a real constant $L > 0$, such that for any $1 \leq i \leq n$ and any $\theta_1, \theta_2$ in the parameter space, we have*

$$\|\nabla f(\theta_1, x_i, y_i) - \nabla f(\theta_2, x_i, y_i)\| \leq L\|\theta_1 - \theta_2\|, \tag{1}$$

*where $\|\cdot\|$ is the Euclidean norm in the space.*

Next, we formally define the **Gradient Trajectory matrix** (GT matrix) mentioned earlier in Section 1.1. Let $g_t^{(l)} = \texttt{vec}(G_t^{(l)})$ be the vectorized version of the gradient matrix at layer $l$ in the neural network at timestamp $t$. For the vanilla SGD optimizer, assume the $T$ gradient updates are $\{g_t^{(l)}\}_{t=0}^{T-1}$. We denote $D_T^{(l)} = [g_0^{(l)}, g_1^{(l)}, \cdots, g_{T-1}^{(l)}]^\top \in \mathbb{R}^{T \times d_l}$ as the GT matrix of layer $l$.

It is worth mentioning that we do not need to instantiate or store $D_T$ in reality, since the existence of analytical form in a local region of parameter space, as stated in the introduction. The consistent gradient structure comes from the recursive back-propagation through layer-wise parameter matrix (Tian, 2020), and is summarized as the property of "reversible network" by Zhao et al. (2024) with different objective functions, where bounded growth in gradient norms can be ensured. Before that, gradient structure has also been researched in Explainable AI (Sixt et al., 2020). In this work, We further analyze the gradient structure of LLM with attention matrix, which is inspired by the analysis of empirical neural tangent kernel (Arora et al., 2019; Ren & Sutherland, 2025), the details will be further discussed in Section 4.2.

To characterize the dimension of the principal subspace, which is spanned by the leading singular vectors of the GT matrix as mentioned earlier, we employ the stable rank (Rudelson & Vershynin,

2007) of the GT matrix as our analytical tool. Being different with Zhao et al. (2024), who employ low-rank decomposition of gradient matrices to reduce the memory costs, our objective is to leverage stable rank to analyze the dimensionality of the principal subspace.

**Definition 1** (Stable Rank of GT Matrix). *The Stable Rank of GT matrix at layer $l$ is defined as:*

$$\text{sr}(D_T^{(l)}) = \frac{\|D_T^{(l)}\|_F^2}{\|D_T^{(l)}\|_2^2} = \frac{\sum_{i=1}^{k} \sigma_i^2(D_T^{(l)})}{\sigma_k^2(D_T^{(l)})}, \tag{2}$$

*where $k$ is the rank of $D_T^{(l)}$, and $0 < \sigma_1(D_T^{(l)}) \leq \cdots \leq \sigma_k(D_T^{(l)})$ denote its sorted singular values.*

In this work, stable rank serves as a scale-invariant proxy for the dimension of the principal space, which is spanned by the main components of GT, *i.e.,* some largest singular vectors of GT matrix. And before we construct a weighted subset, we already have the information (For instance, loss value, gradient, etc.) at current parameter to analyze the stable rank of GT matrix. We will further discuss about the details in Section 4.2.

**Remark 1.** *Stable rank is less sensitive to small perturbations compared to rank, and is invariant to scaling, i.e., $\text{sr}(D) = \text{sr}(D/c), \forall c \in \mathbb{R} \setminus \{0\}$. Moreover, It opens up the possibility for fine-grained analysis, since it is continuous, and differentiable (as both Frobenius and Spectral norms are almost always differentiable).*

## 3 METHOD

In this section, we present our algorithm and provide an overview of the SRS-Sampling approach. For notational simplicity, we use $f_i(\theta)$ hereafter to denote $f(\theta, x_i, y_i)$.

As discussed in the Section 1, many coreset constructions study are based on the concept of "sensitivity". Our approach shares a common foundation with prior work: the use of importance sampling. Intuitively, importance sampling selects data $(x_i, y_i)$ with probability $p_i$, and assigns the weight for its objective function value (loss value) to be $\mu_i = 1/p_i$ when the data is sampled, and $\mu_i = 0$ otherwise. By taking each weighted loss to be a random variable, the above weighted sampling procedure serves as an unbiased estimator for the objective function, *i.e.,* $\mathbb{E}[\mu_i f_i(\theta)] = f_i(\theta)$. Although our method employs importance sampling, similar to "sensitivity" based coreset constructions, it incurs significantly lower computational overhead in determining sampling probability. Specifically, our sampling strategy only requires the per-sample loss values, which are available as by-products of the training process.

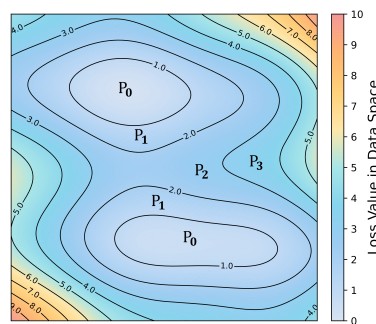

Figure 2: We partition the dataset into different subsets according to loss value computed at $\theta_{\texttt{ini}}$.

To illustrate our sampling strategy, we partition the dataset into regions according to loss values, where each region corresponds to a specific range of losses and each sample is assigned to exactly one region (see Fig. 2). Our goal is to approximate the objective function on the whole dataset using only a subset of weighted data, sampled independently from each region with appropriate probabilities. Formally, let $N = \lceil \log_2 n \rceil$ and set $\hbar = F(\theta_{\texttt{ini}})$, where $\theta_{\texttt{ini}} \in \mathbb{R}^d$ is the initial parameter at which we perform SRS-Sampling for subset selection. Since $f_i(\theta_{\texttt{ini}}) \leq 2^N \hbar$ for any $1 \leq i \leq n$, we partition the input dataset $P = \{(x_1, y_1), (x_2, y_2), \cdots, (x_n, y_n)\}$ into $N + 1$ regions $P = \cup_{j=0}^{N} P_j$ based on the loss values:

$$P_0 = \{(x_i, y_i) \in P \mid f_i(\theta_{\texttt{ini}}) \leq \hbar\}, \tag{3}$$

$$P_j = \{(x_i, y_i) \in P \mid 2^{j-1}\hbar < f_i(\theta_{\texttt{ini}}) \leq 2^j\hbar\}, 1 \leq j \leq N. \tag{4}$$

Note that any exponential partition with a base greater than "1" is applicable to our analysis. The exponential partition scheme leads to a more careful weight allocation to regions with lower loss values, which are of particular interest for gradient descent algorithms. And the choice of base ensures a bounded number of partitions $N$, where a smaller base produces a more refined partition and thus a tighter approximation error bound.

Next, from each non-empty set $P_j$, we uniformly sample a subset $Q_j$ at random, where the sample size $|Q_j|$ will be determined in our theoretical analysis in Section 4. For each sampled data point $(x_i, y_i) \in Q_j$, we assign a weight $\mu_i = \frac{|P_j|}{|Q_j|}$; while all unsampled data points in $P_j \setminus Q_j$ are assigned with weight 0. The resulting weighted subset is specified by the non-zero entries of $\mu_{\mathcal{CS}} = [\mu_1, \mu_2, \cdots, \mu_n]$, and the objective function is approximated by $\tilde{F}(\theta) = \frac{1}{n} \sum_{i=1}^{n} \mu_i f_i(\theta)$. From the construction above, it is easy to verify that $\sum_{i=1}^{n} \mu_i = n$.

This sampling strategy follows the standard construction procedure similar to that of (Chen, 2009; Huang et al., 2021). However, their subset size depends on $d$, which does not scale effectively to deep neural networks. In contrast, we associate the subset size with the stable rank of GT matrix, which captures the dimension of the principal training subspace. We defer the details of theoretical analysis to Section 4, and the construction procedure is shown in Algorithm 1.

---

**Algorithm 1** Stable Rank related Stratified Sampling

---

**Input:** A training dataset $P = \{(x_1, y_1), (x_2, y_2), \cdots, (x_n, y_n)\}$, the Lipschitz constant $L$ as described in Assumption 1, $\theta_{\mathtt{ini}} \in \mathbb{R}^d$, and the radius $R \geq 0$ and $\epsilon \in (0, 1)$.

    1. Let $N = \lceil \log n \rceil$ and $\hbar = F(\theta_{\mathtt{ini}})$; initialize $\mu_{\mathcal{CS}} = [0, 0, \cdots, 0] \in \mathbb{R}^n$.
    2. The set $P$ is partitioned into $N + 1$ layers $\{P_0, \ldots, P_N\}$ as in (3) and (4).
    3. Compute the upper bound of stable rank $\mathrm{sr}(D_T)$ by Algorithm 2.
    4. For each $P_j \neq \emptyset, 0 \leq j \leq N$:
        (a) take a random sample $Q_j$ from $P_j$ uniformly at random, where the size $|Q_j|$ depends on $\mathrm{sr}(D_T)$, the hyper-parameters $\epsilon$, $R$, and $L$ (the exact value will be discussed in our following analysis in Section 4);
        (b) for each sampled data item $(x_i, y_i) \in Q_j$, assign the weight $\mu_i = \frac{|P_j|}{|Q_j|}$;

**Output:** the weight vector $\mu_{\mathcal{CS}} = [\mu_1, \mu_2, \cdots, \mu_n]$.

---

## 4 THEORETICAL ANALYSIS

In this section, we dive into the technical details of the stable rank related sample size $|Q_j|$ for each region. Specifically, we first establish the theoretical guarantees for loss approximation property of Algorithm 1 in Section 4.1. Then in Section 4.2, we analyze the stable rank of the GT matrix defined in Definition 1.

### 4.1 SRS-SAMPLING FOR LOSS APPROXIMATION

Let $\mu_{\mathcal{CS}} = [\mu_1, \mu_2, \cdots, \mu_n]$ be the weight vector returned by running the SRS-Sampling Algorithm 1 at the initial parameter $\theta_{\mathtt{ini}} \in \mathbb{R}^d$, and let $\tilde{F}(\theta) = \frac{1}{n} \sum_{i=1}^{n} \mu_i f_i(\theta)$ be the resulting weighted objective function. Since the gradient norm in training a neural network is typically bounded, we denote $\mathbb{B}(\theta_{\mathtt{ini}}, R)$ as the ball centered at $\theta_{\mathtt{ini}}$ with radius $R \geq 0$. We also define $\mathbb{T} \subseteq \mathbb{B}(\theta_{\mathtt{ini}}, R)$ as a local subspace containing the GT. Recall that $\hbar = F(\theta_{\mathtt{ini}})$ is the average loss computed at $\theta_{\mathtt{ini}}$, and $\mathrm{sr}(D_T)$ is the stable rank of the GT matrix, whose upper bound will be analyzed in Section 4.2. The overall subset size of our proposed SRS-Sampling also depends on two training-specific values: the maximum gradient norm $M := \max\{\|\nabla f_i(\theta_{\mathtt{ini}})\| \mid 1 \leq i \leq n\}$, and the minimum objective value $m := \min_{\theta \in \mathbb{B}(\theta_{\mathtt{ini}}, R)} F(\theta)$.

**Sample size and stable rank**. We now specify the required sample size $\|\mu_{\mathcal{CS}}\|_0$ (the number of non-zero entries of $\mu_{\mathcal{CS}}$ in Algorithm 1). The following Theorem 1 provides a theoretical guarantee for SRS-Sampling Algorithm 1. Specifically, it establishes the relation between required sample size and the success probability to ensure a bounded approximation error of the objective function in $\mathbb{T}$. From Theorem 1, we observe that the sample size depends on the initial vector $\theta_{\mathtt{ini}}$, the stable rank $\mathrm{sr}(D_T)$, and the radius $R$ of the local region. Additionally, the value $m$ is non-increasing with $R$, and the sample size increases with $\mathrm{sr}(D_T)$.

**Theorem 1.** *Given $\epsilon \in (0, 1)$ and the sample size of Algorithm 1 is set to be $\tilde{O}\big(\big(\frac{\hbar + MR + LR^2}{m}\big)^2 \cdot \frac{\mathrm{sr}(D_T)}{\epsilon^2}\big)$ [1]. It satisfies with probability $1 - \frac{1}{n}$ that $\tilde{F}(\theta) \in (1 \pm \epsilon)F(\theta)$, for any $\theta \in \mathbb{T}$.*

**Time complexity**. The weight vector $\mu_{\mathcal{CS}}$, with bounded approximation error specified by Theorem 1, can be obtained in $O(n + \mathtt{Time}_{sr})$ time by running Algorithm 1. Here, $\mathtt{Time}_{sr} = O(kd)$ represents the computational complexity of obtaining the upper bound for $\mathrm{sr}(D_T)$ by Algorithm 2. Typically, we have $k = 2$ for solving two distinct eigen-pairs in Algorithm 2 and $d$ denotes the number of parameters in a neural network. The detailed runtime analysis is deferred to Section B.1.

Next, we provide a proof sketch for Theorem 1, and the complete details are deferred to Section B.1.

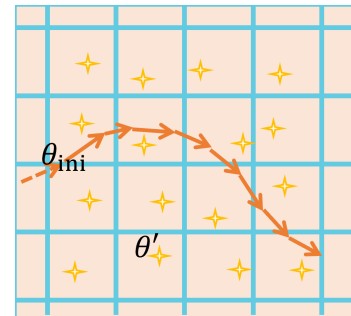

*For a fixed point*. First, we establish the guarantee for an *arbitrarily given* parameter vector $\theta \in \mathbb{T}$. As shown in Lemma 2 (Section B.1), the approximation error $|\tilde{F}(\theta) - F(\theta)|$ can be bounded using Hoeffding's inequality (Hoeffding, 1994).

*A set of points*. Next, we extend the above guarantee from a single point to a set of parameter vectors densely distributed throughout the subspace $\mathbb{T} \subseteq \mathbb{B}(\theta_{\mathtt{ini}}, R)$ (see Fig. 3), which we refer to as "anchors" mentioned in Section 1.1. To do this, we virtually discretize $\mathbb{T}$ into uniform grid cells, with side length $\frac{\epsilon_3 R}{\sqrt{\mathrm{sr}(D_T)}}$. We then arbitrarily pick a point as anchor from each grid cell, and let $V \subseteq \mathbb{T}$ be the set of all anchors. The number of anchors is given by:

$$|V| = O\Big(\big(\frac{2\sqrt{\pi e}}{\epsilon_3}\big)^{\mathrm{sr}(D_T)/\epsilon_3^2}\Big). \tag{5}$$

Figure 3: The subspace (orange plane) is discretized by uniform grid cells, and each grid cell has a randomly sampled anchor.

By applying union bound over all points in $V$, we ensure that the loss approximation holds *simultaneously for all anchors* $\theta' \in V$ with high probability, as shown in Lemma 1. Plug Eq. (5) into Lemma 1, we can see that the required sample size $|Q_j|$ has *linear dependence* on $\mathrm{sr}(D_T)$.

**Lemma 1** (Informal version of Lemma 3). *Suppose $\epsilon_1 \in (0, 1)$ and $\epsilon_2 \geq 0$. We set $|Q_j| = O\big((2^{j-1}\hbar + MR + LR^2)^2\delta^{-2}\log\frac{(N+1)|V|}{\epsilon_1}\big)$ with $\delta = \epsilon_2 2^{j-1}\hbar$ for $j = 0, 1, \cdots, N$. Then for all $\theta' \in V$, we have $|\tilde{F}(\theta') - F(\theta')| \leq \frac{3}{2}\epsilon_2 F(\theta_{\mathtt{ini}})$ with probability at least $1 - \epsilon_1$.*

*The entire GT space*. Finally, leveraging the Lipschitz smoothness of the loss function, we generalize the guarantee from the discrete set of anchors $V$ to any point $\theta$ in the continuous subspace $\mathbb{T}$ containing the GT. Let $\theta'$ be the anchor point of the cell containing $\theta$, as shown by Lemma 4 in the appendix, we can bound the approximation error between $\theta$ and $\theta'$. We ultimately obtain a weighted subset defined by non-zero entries of $\mu_{\mathcal{CS}} = [\mu_1, \mu_2, \cdots, \mu_n]$ in Algorithm 1, and the overall sample size linearly depends on stable rank $\mathrm{sr}(D_T)$ as shown by Theorem 1.

## 4.2 GRADIENT TRAJECTORY IN LOW-DIMENSIONAL SUBSPACE

It remains to present the details about the stable rank of GT matrix $D_T^{(l)}$, which is defined earlier in Definition 1. To do this, we first introduce the consistent gradient structure of reversible network discussed in Section 2.

For a given objective function and a layer parameterized by matrix $W$, one can derive the gradient structure $G = A - BWC$ under vanilla SGD by applying the gradient recursive rules with some straightforward calculations (Zhao et al., 2024). Specifically, for layer $l$, the matrices $A, B, C$ consist of the *recursive Jacobian* $J_l$ after this layer (arising from the chain rule of derivatives) and the *input* $f_{l-1}$ before this layer (since the gradient is input-dependent). Intuitively, the gradient structure measures the gap between the label-related component $A$ and the parameter-related component $BWC$ in the gradient space, where $B$ and $C$ can be seen as the left and right projection matrices for the mapping from the parameter space to the gradient space. The detailed properties of reversible network, objective-function-related structure of the input-dependent matrix $A$ and the Positive Semi-Definite (PSD) matrices $B$ and $C$ can be founded in Definition 2 and Theorem 3 (Section B.2).

---

[1] $\tilde{O}(g) := O\big(g \cdot \mathtt{polylog}\big(\frac{n\hbar MR}{\epsilon m}\big)\big)$.

**Gradient structure for LLM.** We further generalize the above gradient structure to large language model (LLM) under "relatively stable assumption" of attention matrix as mentioned in the introduction, which is similar to the assumption of "stationary" in previous work Li et al. (2023); Tian et al. (2024); Ren & Sutherland (2025). And we shall note that, we do not impose assumptions like frozen projection matrices, or orthogonal embeddings. The intuition behind is straightforward: attention matrix of tokens is a reflection of the inherent property of semantics similarity. Although the hidden representations of tokens evolve throughout the training, the attention matrix of them should change slowly compared with their representations. Similar analysis had been presented in the study of empirical neural tangent kernel (Arora et al., 2019). We refer the interested readers to Theorem 4 (Section B.3) for further analysis about gradient structure of LLM.

**Upper-bound for stable rank**. Note that the aforementioned gradient structure is consistent across all reversible layers, so it is easy to generalize the notation to the gradient of overall reversible network. We omit the layer superscript $l$ for simplicity, and $\mathrm{sr}(D_T)$ can be computed as the summation of stable rank of GT matrix in each layer. Considering a batch of $n$ samples for updates in arbitrary layer, let $g_t = \mathtt{vec}(G_t)$ denote the vectorized gradient matrix, and $\mu_{\mathcal{CS}} = [\mu_1, \mu_2, \cdots, \mu_n]$ be the weights of dataset. The following Theorem 2 establishes the upper-bound of $\mathrm{sr}(D_T)$.

**Theorem 2** (GT resides in a low-dimensional subspace). *Suppose the weighted average gradient follows the structure:* $G_t = \frac{1}{n} \sum_{i=1}^n \mu_i(A_i - B_i W_t C_i)$, *with constant* $A_i$, *PSD matrices* $B_i$ *and* $C_i$. *Consider the vanilla SGD updates of parameter matrix:* $W_t = W_{t-1} + \eta G_{t-1}$. *Let* $S := \frac{1}{n} \sum_{i=1}^n \mu_i(C_i \otimes B_i)$, *where* $\otimes$ *is the Kronecker product, and* $\lambda_1 < \lambda_2$ *are its two smallest distinct eigenvalues.* $g_0^{\parallel}$ *is the component of* $g_0$ *paralleling to the eigenspace corresponding to* $\lambda_1$. *The stable rank* $\mathrm{sr}(D_T)$ *satisfies:*

$$\mathrm{sr}(D_T) \leq 1 + \frac{1 - (1 - \eta\lambda_2)^{2T}}{1 - (1 - \eta\lambda_1)^{2T}} \cdot \frac{\lambda_1(2 - \eta\lambda_1)}{\lambda_2(2 - \eta\lambda_2)} \cdot \frac{\|g_0 - g_0^{\parallel}\|_2^2}{\|g_0^{\parallel}\|_2^2} \tag{6}$$

A proof sketch for Theorem 2 is outlined below, with complete details provided in Section B.4. The derivation involves eigen analysis about the convergence of gradient with operator $(I - \eta S)$, where PSD matrix $S$ depends on sample weights $\mu_{\mathcal{CS}}$ and gradient structure. Intuitively, the operator has the slowest convergence rate of $1 - \eta\lambda_1$ for component $g_0^{\parallel}$ defined in Theorem 2, and its orthogonal complements $g_0 - g_0^{\parallel}$ converges at least at a rate of $1 - \eta\lambda_2$. The result follows by combining with the fact that $\|D_T\|_2^2$ can be lower bounded by the components with slowest convergence rate.

**A small upper bound**. Intuitively, the upper-bound (6) is small, since the eigenspace with the slowest convergence rate concentrates the main components of gradients, and the denominator $\|g_0^{\parallel}\|$ is expected to be dominated. The empirical experiments in Section C.7 further validate that the averaged ratio of $\mathrm{sr}(D_T)$ and $d$ is less than 0.1 for ResNet-18 and ResNet-50.

**Remark 2.** *To take account of updates across all layers, Theorem 5 shows that a multiplicative error will be introduced to the stable rank, which decays superlinear than the learning rate. Please see Section B.4 for more details.*

---

**Algorithm 2** Stable Rank Upper-Bound Computation

---

**Input:** Jacobian $\{J_l\}_0^L$, hidden representations $\{f_l\}_0^L$, layer-wise batch gradient $\{g_0^{(l)} \in \mathbb{R}^{d_l}\}_1^L$ computed at $\theta_{\mathtt{ini}}$, learning rate $\eta$, length of gradient trajectory $T \geq 1$.
**for** layer $l \leftarrow 1$ to $L$ **do**
    Compute $C, B$ based on $J_l, f_{l-1}$ according to the specific gradient structure in Section B.2.
    Compute the two smallest non-zero distinct eigenvalues $\lambda_1 < \lambda_2$ of $S = C \otimes B \in \mathbb{R}^{d_l \times d_l}$.
    Compute $g_0^{\parallel}$ as the component of $g_0^{(l)}$ paralleling to eigenspace corresponding to $\lambda_1$.
    Compute $\mathrm{sr}(D_T^{(l)})$ according to Eq. (6) with $g_0, g_0^{\parallel}, \lambda_1, \lambda_2, \eta, T$.
**end for**
**Output:** the stable rank upper-bound $\mathrm{sr}(D_T) = \sum_{l=1}^L \mathrm{sr}(D_T^{(l)})$.

---

The pseudo code for computing stable rank upper-bound is provided in Algorithm 2. The required gradient $g_0$, Jacobian $J_l$, hidden representation $f_{l-1}$ can be obtained as by-product during training without additional computation. The primary overhead arises from obtaining two smallest distinct

eigen-pairs of $S$. Nevertheless, the process remains efficient for several reasons. First, the computation can be performed layer-wise in parallel, where the dimension of an eigenvector is only the number of elements in a single parameter matrix. Second, only a few eigen-pairs are needed. Third, $S$ is a PSD matrix, many libraries can accelerate the computation based on the symmetric property, for example, *torch.linalg.eigh* (Paszke et al., 2019). The analysis for computational complexity can be found in Section B.1.

## 5 EXPERIMENTS

We implement our proposed SRS-Sampling approach as a sampler. It can be seamlessly integrated into modern distributed training libraries, for example, *Accelerate* (Gugger et al., 2022). With a few lines of code, one can easily integrate our SRS-Sampling approach into the training pipeline like *Transformers* (Wolf et al., 2020). Since our sampling strategy is based on loss value, it is also naturally compatible with training strategies like label smoothing (Szegedy et al., 2016), RandomErasing (Zhong et al., 2017), RandAugment (Cubuk et al., 2019), MixUp/CutMix (Zhang et al., 2018; Yun et al., 2019), large batch size training (You et al., 2017) and exp. moving average (He et al., 2022), low-rank adapter (Hu et al., 2022; Zhao et al., 2024), dataset quantization (Zhou et al., 2023) etc. We conduct all our experiments on *NVIDIA A100-SXM4-80GB* server.

**Datasets and Models.** We validate the effectiveness of proposed SRS-Sampling approach on CIFAR-10, CIFAR-100 (Krizhevsky & Hinton, 2009), ImageNet-1K (Russakovsky et al., 2015), we train from scratch with multiple architectures like ResNet-18, ResNet-50 (He et al., 2016), ViT-Base(MAE) (He et al., 2022) and Swin-Tiny Transformer (Liu et al., 2021). We use AdamW optimizer (Loshchilov & Hutter, 2019) by default in our experiments, SGD and LARS reproduce the similar result (He et al., 2022). Apart from image classification tasks, we also fine-tune Llama-7B (Touvron et al., 2023) on the Alpaca dataset (Taori et al., 2023), and Mistral-7B (Jiang et al., 2023) on the LESS dataset (Xia et al., 2024) using LoRA (Hu et al., 2022). All other details of experiments can be found in Section C.

**Compared approaches.** We set Random* to be the online version of Random, which conducts random selection in each epoch. Many existing coreset approaches for deep learning are based on greedy selection (Sener & Savarese, 2018). They are typically performed offline due to the considerable computational cost of greedy algorithm, meanwhile some of them perform sampling during the training process (Mirzasoleiman et al., 2020; Killamsetty et al., 2021b). We also include data selection methods based on uncertainty, influence, and scores. Among all, InfoBatch (Qin et al., 2024) is a loss-based weighted sampling approach, which is the most similar method to ours. It can be taken as a special case by fixing the number of loss regions in Section 3 to be "2". To be fair, we resample the subset at the beginning of each epoch for all online methods. Detailed descriptions of all compared approaches are provided in Section C.2.

Table 1: Partial results of the Accuracy (%) comparison with state-of-the-art methods on ResNet-18. Since our proposed SRS-Sampling has a dynamic size, we align all methods to have the same iterations counts. Random* denotes online random sampling.

| Dataset | | CIFAR-10 | | | CIFAR-100 | |
|---|---|---|---|---|---|---|
| Subset Ratio % | 30 | 50 | 70 | 30 | 50 | 70 |
| *Offline* Random | $90.2_{\downarrow 5.4}$ | $93.3_{\downarrow 2.3}$ | $94.6_{\downarrow 1.0}$ | $69.7_{\downarrow 8.5}$ | $72.1_{\downarrow 6.1}$ | $73.8_{\downarrow 4.4}$ |
| K-Center (Sener & Savarese, 2018) | $90.9_{\downarrow 4.7}$ | $93.9_{\downarrow 1.7}$ | $94.7_{\downarrow 0.9}$ | $70.2_{\downarrow 8.0}$ | $72.2_{\downarrow 6.0}$ | $74.1_{\downarrow 4.1}$ |
| Least Confidence (Coleman et al., 2020) | $90.3_{\downarrow 5.3}$ | $94.5_{\downarrow 1.1}$ | $95.0_{\downarrow 0.6}$ | $69.8_{\downarrow 8.4}$ | $72.3_{\downarrow 5.9}$ | $74.2_{\downarrow 4.0}$ |
| Influence (Koh & Liang, 2017) | $88.3_{\downarrow 7.3}$ | $91.3_{\downarrow 4.3}$ | $93.1_{\downarrow 2.5}$ | $68.9_{\downarrow 9.5}$ | $72.0_{\downarrow 6.2}$ | $74.4_{\downarrow 3.8}$ |
| EL2N-20 (Paul et al., 2021) | $91.9_{\downarrow 3.7}$ | $\mathbf{95.1}_{\downarrow 0.5}$ | $95.3_{\downarrow 0.3}$ | - | $72.1_{\downarrow 6.1}$ | $77.2_{\downarrow 1.0}$ |
| *Online* Random* | $92.6_{\downarrow 3.0}$ | $93.7_{\downarrow 1.9}$ | $94.7_{\downarrow 0.9}$ | $73.2_{\downarrow 5.0}$ | $75.3_{\downarrow 2.9}$ | $77.3_{\downarrow 0.9}$ |
| Craig (Mirzasoleiman et al., 2020) | $88.4_{\downarrow 7.2}$ | $93.3_{\downarrow 3.3}$ | $94.8_{\downarrow 0.8}$ | $69.7_{\downarrow 8.5}$ | $71.9_{\downarrow 6.3}$ | $74.4_{\downarrow 3.8}$ |
| Glister (Killamsetty et al., 2021b) | $90.9_{\downarrow 4.7}$ | $94.0_{\downarrow 1.6}$ | $95.2_{\downarrow 0.4}$ | $70.4_{\downarrow 7.8}$ | $73.2_{\downarrow 5.0}$ | $74.6_{\downarrow 3.6}$ |
| InfoBatch (Qin et al., 2024) | $94.3_{\downarrow 1.3}$ | $94.9_{\downarrow 0.7}$ | $\mathbf{95.6}_{\uparrow 0.0}$ | $74.6_{\downarrow 3.6}$ | $76.9_{\downarrow 1.3}$ | $77.9_{\downarrow 0.3}$ |
| SRS-Sampling (ours) | $\mathbf{94.6}_{\downarrow 1.0}$ | $\mathbf{95.1}_{\downarrow 0.5}$ | $\mathbf{95.6}_{\uparrow 0.0}$ | $\mathbf{75.7}_{\downarrow 2.5}$ | $\mathbf{77.6}_{\downarrow 0.6}$ | $\mathbf{78.2}_{\uparrow 0.0}$ |
| Overall Dataset | | $95.6_{\pm 0.1}$ | | | $78.2_{\pm 0.1}$ | |

**Results and analysis.** We present a subset of representative methods in Table 1, meanwhile comprehensive comparisons are provided in Table 7. We can gain the following insights from Table 1: (a). *A delicate sampling strategy with theoretical guarantee according to loss regions results in less performance drop.* SRS-Sampling achieves lossless performance at a discard ratio up to 30% (*i.e.,* 70% subset ratio), and is more competitive under small subset ratio, achieving the minimal performance drop of 2.5% on CIFAR-100 and 1% on CIFAR-10 meanwhile reducing 70% of iterations. (b). *Involving stable rank analysis benefits data-efficient training.* Our proposed SRS-Sampling approach efficiently utilizes the loss values and adaptively allocates sample size related to stable rank across training. It outperforms baselines more distinctly on CIFAR-100 than CIFAR-10 (*e.g.,* improvement of 2.3% and 1.4% compared with Random* on 50% data).

Our SRS-Sampling approach can scale to large datasets with *significant reduction in training time meanwhile maintaining competitive performance*. The averaged time consumption and performance across *different optimizers* on ImageNet-1K are shown in Table 2, we only introduce 0.06% runtime overhead, meanwhile achieving lossless performance and reducing overall training time by up to 40% compared with traditional training. Besides the robustness on different optimizers, we also validate its effectiveness across *different architectures* for ImageNet-1K training in Table 3, the results show that we consistently achieve the maximal performance improvement (5.3‰ on average) across baselines. We further validate the effectiveness of SRS-Sampling in Table 4 by *fine-tuning Llama-7B*. We generalize SRS-Sampling to larger-scale fine-tuning scenarios. The results in Table 5 for *fine-tuning Mistral-7B* show a significant advantage of SRS-Sampling over Random*, improving benchmark scores by up to 1.4%. This improvement is achieved using an extremely small subset ratio of 1%, which is a potential application scenario, particularly in large-scale LLM fine-tuning. More details related to LLM fine-tuning can be found in Section C.5.

Table 2: Training time and performance on 60% subset ratio of ImageNet-1K. Results are averaged from SGD/AdamW/LARS optimizers with ResNet-50 for 300 epochs. "Time" is wall clock time; "Total (n*h)" is the total node hour.

| | EL2N-20 | UCB | InfoBatch | SRS-Sampling | Full Data |
|---|---|---|---|---|---|
| Acc (%) | - | $76.3_{\pm0.2}$ | $\underline{76.5_{\pm0.2}}$ | $\mathbf{76.6_{\pm0.2}}$ | $76.6_{\pm0.2}$ |
| Time (h) | 9.7 | 9.7 | 9.7 | 9.7 | 16.2 |
| Overhead (h) | >2.1 | 0.03 | **0.0034** | $\underline{0.0058}$ | 0.0 |
| Total (n*h) | >94.4 | $\underline{77.8}$ | **77.6** | **77.6** | 129.6 |

Table 3: ImageNet-1K results. Three models are trained from scratch for 300 epochs, ViT-Base(MAE) is additionally fine-tuned for 50 epochs following (He et al., 2022), Random* denotes online random sampling.

| Model | Subset Ratio | Random* | InfoBatch | SRS-Sampling |
|---|---|---|---|---|
| ResNet-50 | 60.0% | 76.3 | $76.5_{\uparrow0.2}$ | $\mathbf{76.6_{\uparrow0.3}}$ |
| Swin-T | 80.0% | 80.4 | $80.6_{\uparrow0.2}$ | $\mathbf{80.9_{\uparrow0.5}}$ |
| ViT-B | 80.0% | 82.1 | $82.6_{\uparrow0.5}$ | $\mathbf{82.9_{\uparrow0.8}}$ |

Table 4: Fine-tuning LLaMA-7B on Alpaca. Dataset quantization (DQ) is combined to further decrease computational cost. Subset size is set to 80%. Random* denotes online random sampling. Results of total node time for fine-tuning and Accuracy (%) on four benchmarks are reported.

| Method | Time(m) | BBH | DROP | MMLU | | | | | Human-Eval | Avg. |
|---|---|---|---|---|---|---|---|---|---|---|
| | | | | Overall | Hum. | Other | Soc. | STEM | | |
| DQ+Random* | **115.2** | 33.6 | 27.3 | 34.3 | $\underline{32.6}$ | 37.7 | $\underline{36.6}$ | 30.2 | 10.3 | 26.4 |
| DQ+GREATS | 122.4 | 33.5 | $\underline{27.4}$ | $\underline{34.5}$ | 32.2 | 37.5 | 35.9 | **32.4** | 11.1 | $26.6_{\uparrow0.2}$ |
| DQ+InfoBatch | $\underline{115.3}$ | $\underline{33.7}$ | **27.5** | 33.7 | **33.1** | $\underline{37.9}$ | 33.9 | 30.4 | **11.6** | $26.6_{\uparrow0.2}$ |
| DQ+SRS-Sampling | $\underline{115.3}$ | **33.9** | **27.5** | **34.8** | 31.8 | **38.2** | **39.3** | $\underline{31.7}$ | $\underline{11.5}$ | $\mathbf{26.9_{\uparrow0.5}}$ |

Table 5: Fine-tuning Mistral-7B on LESS. The subset ratio is set to **1%**. Random* denotes online random sampling. We report the total node time for fine-tuning and the zero-shot Accuracy (%) on four benchmarks.

| Method | Time (h) | BBH | HellaSwag | MMLU | | | | | Human-Eval | Avg. |
|---|---|---|---|---|---|---|---|---|---|---|
| | | | | Overall | Hum. | Other | Soc. | STEM | | |
| Random* | 10.31 | 53.4 | 52.3 | 60.4 | 52.6 | 67.5 | 68.4 | 53.2 | 64.3 | 57.6 |
| SRS-Sampling | 10.32 | **55.8** | **53.2** | **61.7** | **54.9** | **68.6** | **69.7** | **53.7** | **65.4** | $\mathbf{59.0_{\uparrow1.4}}$ |

The number of loss strata $N + 1 = \lceil \log_{\texttt{Base}} n \rceil + 1$ in SRS-Sampling depends on the $\texttt{Base}$ of the exponential partition. Here, we perform the sensitivity analysis with respect to the base. We train

ResNet-50 on CIFAR-10 with three independent runs. The size of the dataset is $n = 50000$ and the subset ratio is set to $10\%$. As validated by our experiments in Table 6, the performance is relatively robust to the setting of `Base`, as long as it is not set so large that the number of partitions is too small. Typically, we can set `Base` $\in (1, 5]$. For smaller datasets, a smaller `Base` is preferred to obtain finer partitions.

Table 6: Sensitivity analysis about the base of exponential partitions on CIFAR-10. ResNet-50 is trained for 200 epochs with three independent runs. The subset ratio is set to $10\%$. Random* denotes online random sampling. Test Accuracy (%) is reported.

| Base | 1.4 | 1.6 | 1.8 | 2 | 5 | 10 | Random* |
|---|---|---|---|---|---|---|---|
| #Partitions | 34 | 25 | 20 | 17 | 8 | 6 | 1 |
| Accuracy (%) | $92.7_{\pm 0.3}$ | $93.6_{\pm 0.3}$ | $93.0_{\pm 0.3}$ | $93.4_{\pm 0.3}$ | $92.8_{\pm 0.3}$ | $91.9_{\pm 0.3}$ | $85.6_{\pm 0.3}$ |

Due to space limits, the remaining experiment results can be found in Section C. Specifically, Table 8 (Section C.4) shows that SRS-Sampling has a significant advantage (up to 12.5%) over Random* especially in the early stage of fine-tuning. We perform ablation studies and hyperparameter analysis in Section C.6, which shows the benefit of an adaptive subset size according to the stable rank, and the robustness of SRS-Sampling. More details on SRS-Sampling can be found in Section C.7.

## 6 CONCLUSION

This paper establishes a bridge between the field of data-efficient training and low-dimensional training. Our core idea is to leverage the evolving dimensionality of the Gradient Trajectory (GT) subspace to guide data selection. To this end, we introduce the stable rank of the GT matrix and present a rigorous, fine-grained analysis that determines the necessary subset size for our proposed SRS-Sampling method. Although this paper concentrates on the gradient trajectory, the underlying principles could be extended to other dynamic objects, such as the Hessian trajectory, opening new avenues for algorithm design. We hope this study will stimulate further research at the intersection of data efficiency and training dynamics.

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

## A    OTHER RELATED WORKS

**Low-Rank Analysis for Gradient.**    Several recent studies demonstrate that important machine learning tasks, for example, training deep neural networks and large language models, exhibit a low-rank structure (Li et al., 2018; Zhao et al., 2024). The training of neural networks can be performed explicitly in a low-rank manner through low-rank adapter, without significantly damaging the performance (Hu et al., 2022; Hao et al., 2024; Loeschcke et al., 2024). Besides directly adding an adapter for training, Zhao et al. (2024); Jaiswal et al. (2025); Refael et al. (2025b) establish theoretical analysis about the low-rank structure of gradient matrix. They show that gradient matrix is compressible, such that one could decrease the memory cost of optimizers by storing the low-rank states. Apart from gradient, Refael et al. (2025a) provides low-rank analysis for first-order moment of the optimizer. And Cosson et al. (2023) leverages low-rank structure to mitigate gradient computation cost. Most existing research focus on the low-rank structure of gradient matrix itself, for designing memory-efficient optimization algorithm. Meanwhile we focus on the low-rank property of gradient trajectory matrix, in order to propose a stable rank related stratified sampling approach aware of the evolving dimension of training subspace for efficient utilization of data and training acceleration.

**Coreset for Data-Efficient Training.** Coreset approach (Feldman, 2020) is a representative field of data-efficient training methods. It select a weighted subset such that the performance of machine learning model training on it approximates that on the original dataset. Coreset has already been widely used in optimization problems, such as: clustering (Chen, 2009; Bandyapadhyay et al., 2024), logistic regression (Huggins et al., 2016; Tolochinsky et al., 2022), linear regression (Tukan et al., 2020; Huang et al., 2022a), robust optimization (Ding & Wang, 2020; Huang et al., 2022b), active learning (Coleman et al., 2020; Kim et al., 2020), robust training (Mirzasoleiman et al., 2020; Dolatabadi et al., 2023), continue learning (Borsos et al., 2020; Wang et al., 2022b). For some specific problems in the metric spaces, one could obtain a coreset size better than linear dependence on the dimension of metric spaces (Feldman & Langberg, 2011; Karnin & Liberty, 2019; Baker et al., 2020; Braverman et al., 2021; Cohen-Addad et al., 2021), via techniques like dimension reduction in the input data space (Sohler & Woodruff, 2018; Feng et al., 2021; Woodruff & Yasuda, 2023), or relaxing the worst-case analysis to the average-case analysis (Maalouf et al., 2021). However, for general machine learning problem, the size of the existing coreset approaches still suffers from at least linear dependence on parameter size (Tukan et al., 2020; Mirzasoleiman et al., 2020; Huang et al., 2021; Pooladzandi et al., 2022). A more general research field that encompasses coreset is data selection. Closely related areas include data distillation (Nguyen et al., 2021), data condensation (Zhao & Bilen, 2021; 2023; Kim et al., 2022; Jain et al., 2023), which often involve generating synthetic or condensed datasets, as well as data curation (Evans et al., 2024) and data pruning (Paul et al., 2021; Sorscher et al., 2022; Qin et al., 2024).

## B    PROOFS

### B.1    THEORETICAL GUARANTEE FOR SRS-SAMPLING

We use the following bounds 7 to estimate the range of fluctuations in loss value of each data point lying in different loss regions.

$$f_i(\theta) \in f_i(\theta_{\mathtt{ini}}) \pm \left( \frac{\|\nabla f_i(\theta_{\mathtt{ini}})\|}{R} + \frac{L}{2} \right) R^2. \quad \forall \theta \in \mathbb{T} \subseteq \mathbb{B}(\theta_{\mathtt{ini}}, R), \quad 1 \le i \le n \qquad (7)$$

Bounds 7 could be obtained by combining Taylor expansion and Assumption 1.

**Claim 1.** $\sum_{j=0}^{N} |P_j| 2^j \le 3n$.

*Proof.* By the definition of $P_j$, we have

$$\begin{aligned} 2^j \hbar = \hbar, && \text{if } j = 0; \\ 2^j \hbar \le 2 f_i(\theta_{\mathtt{ini}}), \forall (x_i, y_i) \in P_j, && \text{if } j \ge 1. \end{aligned} \qquad (8)$$

Therefore, $2^j \hbar$ is always no larger than $2 f_i(\theta_{\mathtt{ini}}) + \hbar$ for any $0 \le j \le N$ and any $(x_i, y_i) \in P_j$.

Overall,

$$
\begin{aligned}
\sum_{j=0}^{N} |P_j| 2^j \hbar &= \sum_{j=0}^{N} \sum_{(x_i,y_i)\in P_j} 2^j \hbar \\
&\leq \sum_{j=0}^{N} \sum_{(x_i,y_i)\in P_j} (2 f_i(\theta_{\mathtt{ini}}) + \hbar) \\
&= 2n F(\theta_{\mathtt{ini}}) + n\hbar = 3n\hbar.
\end{aligned}
\tag{9}
$$

Thus the claim $\sum_{j=0}^{N} |P_j| 2^j \leq 3n$ is true. $\qquad\square$

**Lemma 2.** *Given any two numbers $\epsilon_1 \in (0,1)$ and $\epsilon_2 \geq 0$. Note $\delta = \epsilon_2 2^{j-1}\hbar$ for $j = 0, 1, \cdots, N$, if set the sample size of Algorithm 1 to be*

$$
|Q_j| = O\left( (2^{j-1}\hbar + MR + LR^2)^2 \delta^{-2} \log\frac{1}{\epsilon_1} \right),
\tag{10}
$$

*then for any fixed $\theta \in \mathbb{T} \subseteq \mathbb{B}(\theta_{\mathtt{ini}}, R)$,*

$$
\left| \tilde{F}(\theta) - F(\theta) \right| \leq \frac{3}{2}\epsilon_2 F(\theta_{\mathtt{ini}})
\tag{11}
$$

*holds with probability at least $1 - (N+1)\epsilon_1$.*

*Proof.* We first consider $1 \leq j \leq N$. For any given $(x_i, y_i) \in P_j$, we view $f_i(\theta)$ as an independent random variable. Through the partition construction (3) and (4), and the bounds (7), we have

$$
\left.
\begin{aligned}
f_i(\theta) &\geq & 2^{j-1}\hbar - MR - \frac{1}{2}LR^2; \\
f_i(\theta) &\leq & 2^j \hbar + MR + \frac{1}{2}LR^2.
\end{aligned}
\right\}
\tag{12}
$$

Let the sample size $|Q_j| = \lceil \frac{1}{2}(2^{j-1}\hbar + LR^2 + 2MR)^2 \delta^{-2} \log\frac{2}{\epsilon_1} \rceil$. Through the Hoeffding's inequality (Hoeffding, 1994), we know that

$$
\mathtt{Prob}\left[ \left| \frac{1}{|Q_j|} \sum_{(x_i,y_i)\in Q_j} f_i(\theta) - \frac{1}{|P_j|} \sum_{(x_i,y_i)\in P_j} f_i(\theta) \right| \geq \delta \right]
$$

is no larger than $2e^{-\frac{2|Q_j|\delta^2}{(2^{j-1}\hbar + LR^2 + 2MR)^2}} \leq \epsilon_1$.

Now we consider the case $j = 0$. For any given data item $(x_i, y_i) \in P_0$, we have $0 \leq f_i(\theta) \leq \hbar + MR + \frac{1}{2}LR^2$. If letting the sample size $|Q_0| = \lceil \frac{1}{2}(\hbar + \frac{1}{2}LR^2 + MR)^2 \delta^{-2} \log\frac{2}{\epsilon_1} \rceil$, it is easy to verify that the same probability bound also holds.

Combine the two cases above and plugin the value of $\delta$, it holds with probability at most $\epsilon_1$ that

$$
\left| \frac{|P_j|}{|Q_j|} \sum_{(x_i,y_i)\in Q_j} f_i(\theta) - \sum_{(x_i,y_i)\in P_j} f_i(\theta) \right| \geq |P_j|\epsilon_2 2^{j-1}\hbar.
\tag{13}
$$

Recall $\tilde{F}(\theta) = \frac{1}{n} \sum_{i=1}^{n} \mu_i f(\theta, x_i, y_i)$, where for each $(x_i, y_i) \in P_j$, $\mu_i = \frac{|P_j|}{|Q_j|}$ if $(x_i, y_i) \in Q_j$, and $\mu_i = 0$ if $(x_i, y_i) \in P_j \setminus Q_j$. Thus, by taking the union bound of (13) over $0 \leq j \leq N$, we have

$$n\left|\tilde{F}(\theta) - F(\theta)\right|$$

$$= \left| \sum_{j=0}^{N} \frac{|P_j|}{|Q_j|} \sum_{(x_i,y_i) \in Q_j} f_i(\theta) - \sum_{j=0}^{N} \sum_{(x_i,y_i) \in P_j} f_i(\theta) \right|$$

$$\leq \sum_{j=0}^{N} \left| \frac{|P_j|}{|Q_j|} \sum_{(x_i,y_i) \in Q_j} f_i(\theta) - \sum_{(x_i,y_i) \in P_j} f_i(\theta) \right|$$

$$\leq \sum_{j=0}^{N} |P_j| \epsilon_2 2^{j-1} \hbar \tag{14}$$

with probability at least $(1 - \epsilon_1)^{N+1} > 1 - (N+1)\epsilon_1$.

By using Claim 1, (14) can be rewritten as

$$n\left|\tilde{F}(\theta) - F(\theta)\right| \leq \frac{3}{2} \epsilon_2 n F(\theta_{\text{ini}}). \tag{15}$$

So we complete the proof.

$\square$

**Lemma 3.** *Suppose $\epsilon_2 \geq 0$. In the sample size (10) of Lemma 2, we set $\delta = \epsilon_2 2^{j-1} \hbar$ for $j = 0, 1, \cdots, N$, respectively, and replace $\epsilon_1$ in equation 10 by $\frac{\epsilon_1}{(N+1)|V|}$. The following*

$$\left|\tilde{F}(\theta') - F(\theta')\right| \leq \frac{3}{2} \epsilon_2 F(\theta_{\text{ini}}) \tag{16}$$

*holds for all $\theta' \in V$, with probability at least $1 - \epsilon_1$, and $|Q_j|$ in 10 linearly depends on rank $r$.*

*Proof.* Lemma 2 shows that, for a single $\theta$, $|\tilde{F}(\theta) - F(\theta)| > \frac{3}{2} \epsilon_2 F(\theta_{\text{ini}})$ holds with probability at most $(N+1)\epsilon_1$. By replacing $\epsilon_1$ with $\frac{\epsilon_1}{(N+1)|V|}$ into equation 10, it holds with probability at most $\frac{\epsilon_1}{|V|}$ for a single $\theta$. By taking the union bound over all $\theta' \in V$, $|\tilde{F}(\theta') - F(\theta')| \leq \frac{3}{2} \epsilon_2 F(\theta_{\text{ini}})$ holds with probability at least $(1 - \frac{\epsilon_1}{|V|})^{|V|} > 1 - \epsilon_1$.

$\square$

**Lemma 4.** *Define $M' := \max_{1 \leq i \leq n} \max_{\theta \in \mathcal{B}(\theta_{\text{ini}}, R)} \|\nabla f(\theta, x_i, y_i)\|$. Set $\epsilon_2 = \frac{2m\epsilon}{7F(\theta_{\text{ini}})}$ in Lemma 3, where $m = \min_{\theta \in \mathbb{B}(\theta_{\text{ini}}, R)} F(\theta)$. And $\epsilon_3 = \frac{2\epsilon_2 F(\theta_{\text{ini}})}{R\left(\sqrt{M'^2 + 2L\epsilon_2 F(\theta_{\text{ini}})} + M'\right)}$ for equation 5. The following*

$$\left|\tilde{F}(\theta) - F(\theta)\right| \leq \epsilon F(\theta) \tag{17}$$

*holds for all $\theta \in \mathbb{T} \subseteq \mathbb{B}(\theta_{\text{ini}}, R)$, with probability at least $1 - \epsilon_1$.*

*Proof.* By Assumption 1 we immediately know $M' \leq M + LR$. Let $\theta'$ be the anchor point of the cell containing $\theta \in \mathbb{T}$, then we have $\|\theta - \theta'\| \leq \epsilon_3 R$. By using the similar manner of equation 7, for any $1 \leq i \leq n$, we have

$$\left|f_i(\theta) - f_i(\theta')\right| \leq \epsilon_3 M' R + \frac{1}{2} L \epsilon_3^2 R^2. \tag{18}$$

This implies both

$$|F(\theta) - F(\theta')| \text{ and } |\tilde{F}(\theta) - \tilde{F}(\theta')| \quad \leq \quad \epsilon_3 M' R + \frac{1}{2} L \epsilon_3^2 R^2. \tag{19}$$

Combining Lemma 3, bounds 19 and triangle inequality, we obtain

$$
\begin{aligned}
|\tilde{F}(\theta) - F(\theta)| &\leq |\tilde{F}(\theta) - \tilde{F}(\theta')| + |\tilde{F}(\theta') - F(\theta')| + |F(\theta') - F(\theta)| \\
&\leq \frac{3}{2}\epsilon_2 F(\theta_{\mathtt{ini}}) + 2 \times (\epsilon_3 M'R + \frac{1}{2}L\epsilon_3^2 R^2),
\end{aligned}
\tag{20}
$$

By letting $\epsilon_2 = \frac{2m\epsilon}{7F(\theta_{\mathtt{ini}})}$ and $\epsilon_3 = \frac{2\epsilon_2 F(\theta_{\mathtt{ini}})}{R\left(\sqrt{M'^2 + 2L\epsilon_2 F(\theta_{\mathtt{ini}})} + M'\right)}$, we have $|\tilde{F}(\theta) - F(\theta)| \leq \epsilon F(\theta)$ via simple calculations. $\qquad\square$

**Proof of Theorem 1.**

*Proof.* For simplicity, we refer to $\mathrm{sr}(D_T)$ as $r$ in the following. We use Lemma 2 to provide loss approximation for a given $\theta$. Then we use Lemma 3 to generalize the result to the set containing all anchors. Finally, by using the local Lipschitz property, we show that loss approximation guarantee is valid for every $\theta \in \mathbb{T}$ by Lemma 4.

Last, we specify the obtained weighted subset size, *i.e.,* the number of non-zero entries of $\mu_{\mathcal{CS}} = [\mu_1, \mu_2, \cdots, \mu_n]$ returned by Algorithm 1. To guarantee the success probability to be at least $1 - 1/n$, we set $\epsilon_1 = 1/n$. Then we can compute the weighted subset size, *i.e.,* the number of non-zero entries of $\mu_{\mathcal{CS}}$, which equals

$$
\sum_{j=0}^{N} |Q_j| = \tilde{O}\left( \left(\frac{\hbar + MR + LR^2}{m}\right)^2 \cdot \frac{r}{\epsilon^2}\right)
\tag{21}
$$

The above result can be obtained by combining equation 10, with the setting of $\delta$ in Lemma 2, the definition of $\epsilon_2$, the choice of $\epsilon_1$ in Lemma 3, along with equation 5) and definition of $\epsilon_3$.

$\qquad\square$

**Computational complexity.**

*Proof.* The runtime of Algorithm 1 is $O(n + \mathtt{Time}_{sr})$, where $\mathtt{Time}_{sr} = O(kd)$ is the computational complexity for obtaining the upper bound of $\mathrm{sr}(D_T)$ via Algorithm 2 (typically with $k = 2$ and $d$ denotes the number of network parameters).

The linear dependence of $n$ in time complexity is not hard to obtain: our algorithm only requires three passes through the data-wise loss value, which is the by-product of training. It corresponds to the computation of the average loss $\hbar$, the partition of dataset according to 3, 4, and the allocation of weights for samples, respectively.

And $\mathtt{Time}_{sr}$ involves the computation of first $k$ eigen-pairs of the symmetric matrix, which can be computed within $O(d^{(l)}sk)$ time by Lanczos algorithm (Calvetti et al., 1994), where $d^{(l)}$ is the size of symmetric matrix (*i.e.,* the dimension of gradient vector of layer $l$), and $s$ is the average number of nonzero elements in a row. Note that $1 - \eta\lambda_1$ and $1 - \eta\lambda_2$ are the two largest distinct eigenvalues of $I - \eta S$ as shown in the proof of Theorem 2 (B.4). Thus, we typically have $k = 2$ iterations. Since we are computing the eigen-pairs of the Kronecker product as shown in Theorem 2, $d^{(l)}$ can be further decreased by utilizing the basic property of Kronecker product, please see Theorem 4.2.12 of (Horn & Johnson, 1994) for further reading. As a result, $d^{(l)}$ of $S$ can be further decreased to the column size and row size of parameter matrix at layer $l$ (roughly speaking, the hidden size $h$ of $C$ and $B$). Therefore, the computational complexity can be further reduced to $O(k * h^2)$. Typically we have $h^2 = O(d/L)$ for an $L$ layers neural network with total parameter size of $d$. Overall, even if we disregard the parallel-processing capabilities of GPUs for matrix–vector multiplication, and sequentially compute for $L$ layers, the time complexity for Algorithm 2 is only $\mathtt{Time}_{sr} = O(k * L * d/L) = O(kd)$. If we consider the parallel computation based on GPU, the computational complexity can be further reduced. Algorithm 2 is called once in Algorithm 1, so Algorithm 1 has computational complexity $O(n + kd)$ where typically $k = 2$ as mentioned before.

$\qquad\square$

## B.2 REVERSIBLE NETWORK AND GRADIENT STRUCTURE

To ensure no unbounded growth of gradient norms, Tian et al. (2020) introduces a general family of nonlinear networks known as "reversible networks", where each component of such network is reversible.

**Definition 2** (Reversibility (Tian et al., 2020)). *For a layer with expression $\boldsymbol{y} = K(\boldsymbol{x};W)\boldsymbol{x}$, the layer is reversible if the back-propagated gradient at the input $\boldsymbol{x}$ and the output $\boldsymbol{y}$ satisfies $\boldsymbol{g_x} = K(\boldsymbol{x};W)^\top \boldsymbol{g_y}$, where $K(\boldsymbol{x};W)$ depends on the input $\boldsymbol{x}$ and the parameter $W$.*

**Remark 3.** *One should note the difference between "reversible" and "invertible", where "reversible" emphasizes the explicit form of back-propagated gradient, not the explicit form of input given an output. It ensures no unbounded growth in norms (Tian et al., 2020; Jaiswal et al., 2025). And according to the law of chain derivation, it is easy to see the linear combination and the composition of reversible networks are still reversible. Many architectures in real scenarios are reversible, including linear layer (without bias), reversible activations (ReLU, leaky ReLU, polynomials, ...).*

Zhao et al. (2024) shows that for reversible networks $\mathcal{N}(x) = \mathcal{N}_L(\mathcal{N}_{L-1}(\cdots\mathcal{N}_1(x))) = K_L(x)K_{L-1}(x)\cdots K_1(x)x$, it can be decomposed as $\mathcal{N}(x) = J_l W_l f_{l-1}$, where $f_{l-1} := \mathcal{N}_l(\mathcal{N}_{l-1}(\cdots\mathcal{N}_1(x)))$ and $J_l := K_L(x)\cdots K_{l+1}(x)x$. For vanilla SGD parameter update $W_t = W_{t-1} + \eta G_{t-1}$, the gradient has the following structure:

$$G_t = A_t - B_t W_t C_t \tag{22}$$

where matrix $A_t$ is input-dependent, and matrices $B_t, C_t$ are Positive Semi-Definite (PSD).

**Theorem 3** (Gradient Form of reversible models (Zhao et al., 2024)). *Consider a chained reversible neural network $\mathcal{N}(x) := \mathcal{N}_L(\mathcal{N}_{L-1}(\ldots\mathcal{N}_1(x)))$ and define $J_l := \text{Jacobian}(\mathcal{N}_L)\ldots\text{Jacobian}(\mathcal{N}_{l+1})$ and $f_l := \mathcal{N}_l(\ldots\mathcal{N}_1(x))$. Then the weight matrix $W_l$ at layer $l$ has gradient $G_l$ in the following form for batch size 1:*

*Let $P_K^\perp = I - \frac{1}{K}11^\top$ be the zero-mean PSD projection matrix. For $K$-way logsoftmax loss $\phi(y; f_L) := -\log\left(\frac{\exp(y^\top f_L)}{1^\top \exp(f_L)}\right)$ with small logits $\|P_K^\perp f_L\|_\infty \ll \sqrt{K}$:*

$$G_l = \underbrace{J_l P_K^\perp y f_{l-1}^\top}_{A} - \underbrace{\gamma J_l^\top P_K^\perp J_l}_{B} W_l \underbrace{f_{l-1} f_{l-1}^\top / K}_{C} \tag{23}$$

*where $\gamma \approx 1$ and $y$ is a data label with $y^\top 1 = 1$.*

## B.3 GRADIENT STRUCTURE FOR TRANSFORMER ARCHITECTURE

Our proof proceeds in two parts. First, we show that the backpropagation through the overall network architecture is similar to that of a reversible network under our stability assumption. Second, we perform a detailed analysis for each parameter matrix within the attention block $(W_V, W_Q, W_K)$ to show they all conform to the $A - BWC$ structure.

To begin with, the attention matrix measures the normalized similarity of token representations within a given prefix of sentence. It should change slowly during training compared with the evolution of token representations, since such similarity is the inherent property of token semantics. Under such relatively stable assumption of attention matrix, the back-propagation structure of gradient is similar to that of a reversible network defined in Eq. (22), and the analysis in Theorem 2 can still be generalized to large language model.

**Theorem 4.** *Assuming the attention matrix is relatively stable with respect to the evolution of token representations, then the transformer architecture with logsoftmax loss is approximately reversible following the Definition 2, and the gradients follow the structure $G_t = A_t - B_t W_t C_t$, where $A_t$ is input-dependent, $B_t, C_t$ are Positive Semi-Definite (PSD).*

*Proof.* Typically speaking, the transformer architecture in large language model consisting of blocks of attention layer Eq. (24) and multi-layer perception (MLP) Eq. (25):

$$Y_l = W_{l,V} X_l f_{\texttt{softmax}}\left(X_l^\top W_{l,K}^\top W_{l,Q} X_l\right) = W_{l,V} X_l C_l \tag{24}$$

$$X_{l+1} = W_{l,o} f_{\texttt{act}}\left(W_{l,h} Y_l\right) = K_l Y_l \tag{25}$$

where $Y_l$ is the output of attention layer $l$ and the input of MLP layer $l$, $X_l$ is the input of attention layer $l$ and the output of MLP layer $l - 1$, each column of $X_l$ ($Y_l$) stands for a token representation. $f_{\texttt{softmax}}$ is the softmax function applied separately to each column to obtain the normalized similarity of keys given a specific query. $f_{\texttt{act}}$ is the activation function. We note $C_l := f_{\texttt{softmax}}\left(X_l^\top W_{l,K}^\top W_{l,Q} X_l\right)$ and $K_l$ being the input-weight-dependent matrix.

In the following, we will use $X_l$, $f_l$ interchangeably, since $X_l$ can be seen as the output of a transformer block containing attention layer and MLP layer, which plays the same role as layer output $f_l$, i.e., $X_l = f_l = J_{l'} W_{l'} f_{l'-1}, \forall 0 < l' < l$, for analyzing the gradient of $W_{l'}$.

Note that:

$$X_{l+1} = K_l W_{l,V} X_l C_l = K_l W_{l,V} K_{l-1} W_{l-1,V} X_{l-1} C_{l-1} C_l = \cdots = [KW_V]_{l:0} X_0 C_{0:l} \quad (26)$$

where $[KW_V]_{l:0} := K_l W_{l,V} K_{l-1} W_{l-1,V} \cdots K_0 W_{0,}$, and in the following we will use such notation $[\cdot]_{a:b}$ to identify the matrix multiplication with orderly subscripts from $a$ to $b$.

Since the attention matrix $C_l$ is relatively stable with respect to the evolution of token representations $X_l$, i.e., $\frac{\partial C_l}{\partial X_l} \approx 0$, $\forall l \in \{0, \cdots, L - 1\}$, it is easy to see that:

$$\mathrm{d}X_L = [KW_V]_{L:l}\mathrm{d}X_l C_{l:L} \qquad + \qquad \text{terms not related to } \mathrm{d}X_l \quad (27)$$

From Eq. (27), one can see that the gradient back-propagation across layers is the same with a reversible network (by taking $[KW_V]_{L:l+1}$ as Jacobian after $X_{l+1}$ and $C_{l+1:L}$ as input before $X_{l+1}$). Then by following the analysis in Zhao et al. (2024), the gradients of output weight $W_{l,o}$ and hidden weight $W_{l,h}$ in MLP layer (25) will follow structure Eq. (22) for reversible activation function $f_{\texttt{act}}$.

To this end, we have shown that attention layer is approximately reversible, then we can focus our gradient analysis to the query/key/value matrix $W_{l,Q}, W_{l,K}, W_{l,V}$ in the attention layer Eq. (24).

For a column vector $z$ with dimension $n_h$, since $f_{\texttt{softmax}}(z) = \frac{1}{n_h}1 + \frac{1}{n_h}(I - \frac{1}{n_h}11^\top)z + \mathcal{O}(z^2)$, by noting $P_{n_h}^\parallel = \frac{1}{n_h}11^\top, P_{n_h}^\perp = I - \frac{1}{n_h}11^\top$, we have:

$$C_l \approx P_{n_h}^\parallel + P_{n_h}^\perp X_l^\top W_{l,K}^\top W_{l,Q} X_l / n_h \quad (28)$$

Similarly, for the K-way logsoftmax loss $\varphi(y; f) = -\log\left(\frac{\exp(y^\top f)}{1^\top \exp(f)}\right)$ with label $y$ satisfying $y^\top 1 = 1$. The Lemma B.2 in Zhao et al. (2024) show that:

$$-\mathrm{d}\varphi = y^\top \mathrm{d}\hat{f} - \gamma \hat{f}^\top \mathrm{d}\hat{f}/K + \mathcal{O}(\hat{f}^2/K)\mathrm{d}\hat{f} \quad (29)$$

where $\gamma(y, f) \approx 1$, and $\hat{f} = P_K^\perp f$ with $P_K^\perp = I - \frac{1}{K}11^\top$.

In the following, we are going to show that by decomposing model output $f_L = X_L = J_l W_l f_{l-1} + R_\neg$, the gradient $G_{l,V}, G_{l,K}^\top, G_{L,Q}$ corresponding to weight matrix $W_{l,V}, W_{l,K}^\top, W_{L,Q}$ follows the structure $J_l^\top P_K^\perp y f_{l-1}^\top + R'_\neg - \gamma J_l^\top P_K^\perp J_l W_l f_{l-1} f_{l-1}^\top/K$, where $R_\neg$ is not related to the parameter $W_l$ and $R'_\neg$ is not related to $\mathrm{d}W_l$. That is, the gradient structure Eq. (22) is still valid for transformer model.

**For value matrix $W_{l,V}$.**

Note that:

$$f_L = [KW_V]_{L:l+1} K_l W_{l,V} X_l C_{l:L} \quad (30)$$

So we have:

$$J_l = [KW_V]_{L:l+1} K_l \quad (31)$$
$$f_{l-1} = X_l C_{l:L} \quad (32)$$

Then according to Eq. (29):

$$-\mathrm{d}\varphi = \texttt{tr}\left(f_{l-1}^\top \mathrm{d}W_{l,V}^\top J_l^\top (P_K^\perp y - \gamma P_K^\perp J_l W_{l,V} f_{l-1}/K)\right) \quad (33)$$
$$= \texttt{tr}\left(\mathrm{d}W_{l,V}^\top J_l^\top (P_K^\perp y - \gamma P_K^\perp J_l W_{l,V} f_{l-1}/K) f_{l-1}^\top\right) \quad (34)$$

So the $G_{l,V}$ is:

$$G_{l,V} = \underbrace{J_l^\top P_K^\perp y f_{l-1}^\top}_{A} - \underbrace{\gamma J_l^\top P_K^\perp J_l}_{B} W_{l,V} \underbrace{f_{l-1} f_{l-1}^\top / K}_{C} \tag{35}$$

where $J_l, f_{l-1}$ are defined in Eqs. (31) and (32).

**For query matrix $W_{l,Q}$.**

Plug in Eq. (28) to Eq. (26):

$$f_L = [KW_V]_{L:l} X_l (P_{n_h}^\| + P_{n_h}^\perp X_l^\top W_{l,K}^\top W_{l,Q} X_l / n_h) C_{l+1:L} \tag{36}$$

$$= [KW_V]_{L:l} X_l P_{n_h}^\| C_{l+1:L} + [KW_V]_{L:l} X_l P_{n_h}^\perp X_l^\top W_{l,K}^\top W_{l,Q} X_l C_{l+1:L} / n_h \tag{37}$$

So we have:

$$J_l = [KW_V]_{L:l} X_l P_{n_h}^\perp X_l^\top W_{l,K}^\top \tag{38}$$

$$f_{l-1} = X_l C_{l+1:L} / n_h \tag{39}$$

$$R_\neg = [KW_V]_{L:l} X_l P_{n_h}^\| C_{l+1:L} \tag{40}$$

Then according to Eq. (29):

$$-d\varphi = \mathtt{tr}\left(f_{l-1}^\top dW_{l,Q}^\top J_l^\top (P_K^\perp y - \gamma P_K^\perp f_L / K)\right) \tag{41}$$

$$= \mathtt{tr}\left(dW_{l,Q}^\top J_l^\top (P_K^\perp y - \gamma P_K^\perp f_L / K) f_{l-1}^\top\right) \tag{42}$$

So the $G_{l,Q}$ is:

$$G_{l,Q} = J_l^\top P_K^\perp y f_{l-1}^\top - \gamma J_l^\top P_K^\perp (J_l W_{l,Q} f_{l-1} + R_\neg) / K) f_{l-1}^\top \tag{43}$$

$$= \underbrace{J_l^\top P_K^\perp (y - \gamma R_\neg / K) f_{l-1}^\top}_{A} - \underbrace{\gamma J_l^\top P_K^\perp J_l}_{B} W_{l,Q} \underbrace{f_{l-1} f_{l-1}^\top / K}_{C} \tag{44}$$

where $J_l, f_{l-1}, R_\neg$ are defined in Eqs. (38) to (40).

**For key matrix $W_{l,K}$.** From Eq. (37) we can see $f_L = J_l W_{l,K}^\top f_{l-1} + R_\neg$ where:

$$J_l = [KW_V]_{L:l} X_l P_{n_h}^\perp X_l^\top \tag{45}$$

$$f_{l-1} = W_{l,Q} X_l C_{l+1:L} / n_h \tag{46}$$

$$R_\neg = [KW_V]_{L:l} X_l P_{n_h}^\| C_{l+1:L} \tag{47}$$

Then according to Eq. (29):

$$-d\varphi = \mathtt{tr}\left(y^\top P_K^\perp df_L - \gamma f_L^\top P_K^\perp df_L / K\right) \tag{48}$$

$$= \mathtt{tr}\left((y^\top P_K^\perp - \gamma f_L^\top P_K^\perp / K) J_l dW_{l,K}^\top f_{l-1}\right) \tag{49}$$

$$= \mathtt{tr}\left(dW_{l,K}^\top f_{l-1} (y^\top P_K^\perp - \gamma f_L^\top P_K^\perp / K) J_l\right) \tag{50}$$

So the $G_{l,K}$ is:

$$G_{l,K} = f_{l-1} y^\top P_K^\perp J_l - \gamma f_{l-1} (R_\neg^\top + f_{l-1}^\top W_{l,K} J_l^\top) P_K^\perp J_l / K \tag{51}$$

$$= \underbrace{f_{l-1} (y^\top - \gamma R_\neg^\top / K) P_K^\perp J_l}_{A} - \underbrace{\gamma f_{l-1} f_{l-1}^\top}_{B} W_{l,K} \underbrace{J_l^\top P_K^\perp J_l / K}_{C} \tag{52}$$

where $J_l, f_{l-1}, R_\neg$ are defined in Eqs. (45) to (47).

To this end, we have shown that $G_{l,Q}, G_{l,K}, G_{l,V}$ in attention layer (24) also follow the gradient structure defined in Eq. (22). $\qquad\square$

### B.4 GRADIENT TRAJECTORY IN LOW-DIMENSIONAL SUBSPACE

**Proof of Theorem 2.**

*Proof.* We can recursively express $G_t$ as:

$$G_t = \frac{1}{n} \sum_{i=1}^{n} \mu_i \left( A_i - B_i W_t C_i \right) = \frac{1}{n} \sum_{i=1}^{n} \mu_i A_i - \mu_i B_i \left( W_{t-1} + \eta G_{t-1} \right) C_i$$

$$= G_{t-1} - \frac{\eta}{n} \sum_{i=1}^{n} \mu_i B_i G_{t-1} C_i \tag{53}$$

Let $S := \frac{1}{n} \sum_{i=1}^{n} \mu_i (C_i \otimes B_i) \in \mathbb{R}^{d \times d}$ to be a PSD matrix (since $B_i, C_i$ are PSD matrix), where $\otimes$ is the Kronecker product, and $g_t := \text{vec}(G_t) \in \mathbb{R}^d$ be a vectorized version of the gradient $G_t \in \mathbb{R}^d$. Using $\text{vec}(BWC) = (C^\top \otimes B)\text{vec}(W)$, we have:

$$g_t = (I - \eta S) g_{t-1} \tag{54}$$

Note that Eq. (54) is valid for each layer in a reversible network. Next, we can bound the stable rank of gradient trajectory:

$$\text{sr}(D_T) = \frac{\|D_T\|_F^2}{\|D_T\|_2^2}. \tag{55}$$

We can decompose $g_0$ into two components, one is $g_0^{\parallel}$ lying in the $\kappa_1$-dimensional eigenspace $\mathcal{V}_1 \subset \mathbb{R}^d$ corresponding to $\lambda_1$, where $\kappa_1$ is the multiplicity of smallest eigenvalue $\lambda_1$, the other is $g_0^{\perp} := g_0 - g_0^{\parallel}$. Since $\mathcal{V}_1$ and its orthogonal complements are invariant subspaces under $S$, we have the following upper bound on $\|D_T\|_F^2$:

$$\|D_T\|_F^2 = \sum_{t=0}^{T-1} \|g_t\|_2^2 = \sum_{t=0}^{T-1} \|(I - \eta S)^t g_0\|_2^2 \tag{56}$$

$$= \sum_{t=0}^{T-1} \left\{ \|(I - \eta S)^t g_0^{\perp}\|_2^2 + \|(I - \eta S)^t g_0^{\parallel}\|_2^2 \right\} \tag{57}$$

$$\leq \sum_{t=0}^{T-1} \left\{ (1 - \eta \lambda_2)^{2t} \|g_0^{\perp}\|_2^2 + (1 - \eta \lambda_1)^{2t} \|g_0^{\parallel}\|_2^2 \right\} \tag{58}$$

$$= \frac{1 - (1 - \eta \lambda_2)^{2T}}{\eta \lambda_2 (2 - \eta \lambda_2)} \|g_0^{\perp}\|_2^2 + \frac{1 - (1 - \eta \lambda_1)^{2T}}{\eta \lambda_1 (2 - \eta \lambda_1)} \|g_0^{\parallel}\|_2^2 \tag{59}$$

where inequality. (58) uses the fact that $\lambda_1$ is the eigenvalue of $\mathcal{V}_1$, and $\lambda_2$ is the smallest eigenvalue of the complement space of $\mathcal{V}_1$.

Notice that $\|D_T\|_2^2$ is the maximum eigenvalue of $D_T^\top D_T$, where $D_T^\top D_T = \sum_{t=0}^{T-1} \left\{ g_t g_t^\top \right\} = \sum_{t=0}^{T-1} \left\{ [(I - \eta S)^t g_0][(I - \eta S)^t g_0]^\top \right\}$, so the lower bound on $\|D_T\|_2^2$ is:

$$\|D_T\|_2^2 = \max_{\|v\|=1} v^\top D_T^\top D_T v \tag{60}$$

$$= \max_{\|v\|=1} \sum_{t=0}^{T-1} \left( v^\top (I - \eta S)^t g_0 \right)^2 \tag{61}$$

$$\geq \max_{\|v\|=1, v \in \mathcal{V}_1} \sum_{t=0}^{T-1} (1 - \eta \lambda_1)^{2t} (v^\top g_0)^2 \tag{62}$$

$$= \frac{1 - (1 - \eta \lambda_1)^{2T}}{\eta \lambda_1 (2 - \eta \lambda_1)} \|g_0^{\parallel}\|_2^2 \tag{63}$$

where inequality. (62) uses the fact that orthogonal complement space of $\mathcal{V}_1$ is invariant under $S$ and $v^\top g_0^{\perp} = 0$ for any $v \in \mathcal{V}_1$.

Combine the Eqs. (59) and (63) to finish the proof.

$\square$

**Theorem 5** (Low-rank GT Matrix considering influence across layers). *Assuming $A_t, B_t, C_t$ in the gradient structure Eq. (22) have $L_A, L_B, L_C$ continuity with respect to $W_t$, and bounded parameter $\|W_t\| \leq D$. Let $\lambda_{\texttt{max}}$ be the maximum eigenvalue of $S_0 = C_0 \otimes B_0$, where $\otimes$ is the Kronecker product, and $\lambda_1 < \lambda_2$ are its two smallest distinct eigenvalues. $g_0^{\|}$ is the component of $g_0$ paralleling to eigenspace corresponding to $\lambda_1$. Set $\varepsilon_e = \eta(L_A + 2L_B L_C D^2)/(1 - \eta\lambda_{\texttt{max}})$, for small enough learning rate $\eta$ of vanilla SGD, we can assure that $\varepsilon_e \in \mathcal{E} \subset (0, 1)$. The joint effect across all layers at previous step will introduce a multiplicative error of $(1 + O(\varepsilon_e))$ to the stable rank of gradient trajectory with convergence operator $(I - \eta S_0)$. That is:*

$$\mathrm{sr}(D_T) \leq (1 + O(\varepsilon_e)) \cdot \left( 1 + \frac{1 - (1 - \eta\lambda_2)^{2T}}{1 - (1 - \eta\lambda_1)^{2T}} \cdot \frac{\lambda_1(2 - \eta\lambda_1)}{\lambda_2(2 - \eta\lambda_2)} \cdot \frac{\|g_0 - g_0^{\|}\|_2^2}{\|g_0^{\|}\|_2^2} \right) \quad (64)$$

*Proof.* Let $g_0^{\perp} := g_0 - g_0^{\|}$, $w_t = \texttt{vec}(W_t)$, $a_t = \texttt{vec}(A_t)$, $S_t = C_t \otimes B_t$. The given gradient trajectory $\hat{D}_T$ without considering the updates of other layers follows $\hat{g}_t := (I - \eta S_0)\hat{g}_{t-1}$, we now calibrate the trajectory to reflects the joint effects of updates across all layers. For the perturbation introduced by updates of other layers at timestep $t-1$ such that $a_t \neq a_{t-1}$, $S_t \neq S_{t-1}$, we consider $g_t := a_t - S_t w_t = \hat{g}_t + e_t$, where $e_t = a_t - a_{t-1} - (S_t - S_{t-1})w_t$ for $t \in \{1, \cdots, T-1\}$, and $g_0 = \hat{g}_0$.

Next, we bound the perturbation $e_t$. Note that:

$$\|a_t - a_{t-1}\|_2 = \|A_t - A_{t-1}\|_F \leq L_A \|W_t - W_{t-1}\|_F = \eta L_A \|\hat{G}_{t-1}\|_F \quad (65)$$

$$\|(B_t - B_{t-1})W_t C_t\|_F \leq L_B \|W_t - W_{t-1}\|_F \|W_t\|_F \|C_t\|_F = \eta L_B L_C D^2 \|\hat{G}_{t-1}\|_F \quad (66)$$

$$\|B_{t-1}W_t(C_t - C_{t-1})\|_F \leq L_C \|B_{t-1}\|_F \|W_t\|_F \|W_t - W_{t-1}\|_F \leq \eta L_B L_C D^2 \|\hat{G}_{t-1}\|_F \quad (67)$$

Combing Eqs. (66) and (67), we have:

$$\|(S_t - S_{t-1})w_t\|_2 = \|B_t W_t C_t - B_{t-1}W_t C_{t-1}\|_F \quad (68)$$

$$\leq \|(B_t - B_{t-1})W_t C_t\|_F + \|B_{t-1}W_t(C_t - C_{t-1})\|_F \quad (69)$$

$$\leq 2\eta L_B L_C D^2 \|\hat{G}_{t-1}\|_F \quad (70)$$

Combing Eqs. (65) and (70) and using the fact that $\|\hat{g}_{t-1}\|_2 = \|\hat{G}_{t-1}\|_F$, we have the upper bound on $\|e_t\|_2$:

$$\|e_t\|_2 \leq \|a_t - a_{t-1}\|_2 + \|(S_t - S_{t-1})w_t\|_2 \quad (71)$$

$$\leq \eta(L_A + 2L_B L_C D^2)\|\hat{g}_{t-1}\|_2 \quad (72)$$

$$\leq \|\hat{g}_t\|_2 \cdot \eta(L_A + 2L_B L_C D^2)/(1 - \eta\lambda_{\texttt{max}}) \quad (73)$$

Finally, set $\varepsilon_e = \eta(L_A + 2L_B L_C D^2)/(1 - \eta\lambda_{\texttt{max}})$, and obtain $\|e_t\|_2 \leq \varepsilon_e \|\hat{g}_t\|_2$, then the gradient $g_t$ considering the influence of updates across layers can be bounded by:

$$\|g_t\|_2 \in \|(1 \pm \varepsilon_e)\hat{g}_t\|_2 \quad (74)$$

Let $D_T = \hat{D}_T + E_T$, for small enough $\varepsilon_e$, we have:

$$\|D_T\|_F^2 = \sum_{t=0}^{T-1} \|g_t\|_2^2 \leq (1 + \varepsilon_e)^2 \sum_{t=0}^{T-1} \|\hat{g}_t\|_2^2 = (1 + \varepsilon_e)^2 \|\hat{D}_T\|_F^2 \quad (75)$$

$$\|D_T\|_2 \geq \|\hat{D}_T\|_2 - \|E_T\|_2 \geq \|\hat{D}_T\|_2 - \|E_T\|_F \geq \|\hat{D}_T\|_2 - \varepsilon_e \|\hat{D}_T\|_F \quad (76)$$

Plug in to the definition of stable rank, and let $r' := \mathrm{sr}(\hat{D}_T)$, when $\varepsilon_e \ll 1$, we have:

$$\mathrm{sr}(D_T) \leq \frac{(1 + \varepsilon_e)^2 r'}{(1 - \varepsilon_e\sqrt{r'})^2} \quad (77)$$

$$\lesssim (1 + 2\varepsilon_e(1 + \sqrt{r'}))r' \quad (78)$$

$$\lesssim (1 + O(\varepsilon_e))r' \quad (79)$$

where Eq. (78) ignores high order term containing $\varepsilon_e^2$, since $\varepsilon_e \to 0$ when $\eta \to 0$. To this end, we have obtained the multiplicative error of $(1 + O(\varepsilon_e))$, which has a superlinear decrease compared to the learning rate $\eta$.

$\square$

## B.5 Subspace Dimension and Covering Number

Our goal is to construct a weighted subset that provides a uniform loss approximation guarantee over the space in which the local parameter trajectory $[\theta_1, \ldots, \theta_T]$ resides. Our approach is to first identify a low-dimensional subspace that contains the trajectory, and then build a weighted subset for a fine-grained discretization (an $\epsilon'$-net) of that subspace. The following lemma establishes the crucial link between the stable rank of the GT matrix and the dimension of this subspace, which determines the size of the required discretization in Lemma 3.

**Lemma 5** (Trajectory Subspace Dimension via Stable Rank). *Let the local parameter trajectory be $[\theta_1, \ldots, \theta_T]$, generated from gradients whose matrix is $D_T \in \mathbb{R}^{T \times d}$. For a given precision $\epsilon > 0$, there exists a $k$-dimensional affine subspace $\mathcal{A}_k$ that contains the entire trajectory within a sufficiently small neighborhood, provided the dimension $k$ satisfies*

$$k = O\left(\frac{\mathrm{sr}(D_T)}{\epsilon^2}\right). \tag{80}$$

*Consequently, the portion of this subspace within the local ball $\mathbb{B}(\theta_{\mathtt{ini}}, R)$ can be covered by an $\epsilon'$-net of size $|V| = \left(O\left(\frac{R}{\epsilon'}\right)\right)^k$, where the exponent $k$ depends directly on the stable rank and the desired precision $\epsilon$.*

*Proof.* The parameter trajectory is defined by the cumulative sum of gradients: $\theta_t = \theta_{\mathtt{ini}} - \eta \sum_{i=1}^{t} g_i$. Let $\mathcal{A}_k = \theta_{\mathtt{ini}} + S_k$ be the affine subspace, where $S_k$ is the linear subspace spanned by the top $k$ right singular vectors of the gradient matrix $D_T$. Let $\sigma_1(D_T^{(l)}) \geq \cdots \geq \sigma_k(D_T^{(l)}) \geq \cdots$. The deviation of any point on the trajectory from this subspace is bounded by the cumulative projection error of the gradients:

$$\mathrm{dist}(\theta_t, \mathcal{A}_k) = \|(\theta_t - \theta_{\mathtt{ini}}) - P_{S_k}(\theta_t - \theta_{\mathtt{ini}})\|_2 = \eta \left\|\sum_{i=1}^{t}(g_i - P_{S_k}(g_i))\right\|_2. \tag{81}$$

To ensure this deviation is small for the entire trajectory without an explicit dependence on the trajectory length $T$, we require the per-step gradient approximation error to be controlled relative to the overall scale of the gradients. This implies that the $(k+1)$-th singular value of the gradient matrix, $\sigma_{k+1}(D_T)$, which serves as a measure of the worst-case projection error, must be small. Since $\max_{i \in [T]} \|g_i\|_2 \leq \|D_T\|_2 = \sigma_1(D_T)$, we select $k$ such that

$$\sigma_{k+1}(D_T) \leq \epsilon \, \sigma_1(D_T). \tag{82}$$

Then the cumulative projection error in equation 81 can be bounded. By the triangle inequality and then the Cauchy-Schwarz inequality, we have:

$$\mathrm{dist}(\theta_t, \mathcal{A}_k) \leq \eta \sum_{i=1}^{t} \|g_i - P_{S_k}(g_i)\|_2$$

$$\leq \eta\sqrt{t}\left(\sum_{i=1}^{t}\|g_i - P_{S_k}(g_i)\|_2^2\right)^{1/2} \leq \eta\sqrt{T}\left(\sum_{i=1}^{T}\|g_i - P_{S_k}(g_i)\|_2^2\right)^{1/2}$$

$$= \eta\sqrt{T}\|D_T - D_T P_{S_k}\|_F. \tag{83}$$

The Frobenius norm of the error matrix is bounded by its operator norm: $\|D_T - D_T P_{S_k}\|_F^2 \leq T \cdot \|D_T - D_T P_{S_k}\|_2^2 = T \cdot \sigma_{k+1}^2(D_T)$. Plugging this in and using our condition from equation 82:

$$\mathrm{dist}(\theta_t, \mathcal{A}_k) \leq \eta\sqrt{T}\left(T \cdot \sigma_{k+1}^2(D_T)\right)^{1/2} = \eta T \sigma_{k+1}(D_T) \leq \eta T \epsilon \sigma_1(D_T). \tag{84}$$

Assuming the trajectory radius $R$ is on the order of $\eta T \sigma_1(D_T)$, this deviation is bounded by $O(R\epsilon)$. For simplicity, we write it as $R\epsilon$.

We can relate this condition to the required dimension $k$ using the definition of stable rank. A standard inequality states that

$$\sigma_{k+1}^2 \leq \frac{\|D_T\|_F^2}{k+1} = \frac{\text{sr}(D_T) \cdot \sigma_1^2(D_T)}{k+1}. \tag{85}$$

To satisfy our condition "$\sigma_{k+1} \leq \epsilon\sigma_1$" in equation 82, we need the following inequalities:

$$\frac{\text{sr}(D_T) \cdot \sigma_1^2(D_T)}{k+1} \leq \epsilon^2 \sigma_1^2(D_T). \tag{86}$$

Solving for $k$, we can take $k = O\left(\frac{\text{sr}(D_T)}{\epsilon^2}\right)$.

Finally, the size of an $\epsilon'$-net required to cover the intersection of the $k$-dimensional subspace $\mathcal{A}_k$ with the local ball $\mathbb{B}(\theta_{\texttt{ini}}, R)$ is given by the standard covering number for a $k$-dimensional Euclidean ball:

$$|V| \leq (1 + \frac{2R}{\epsilon'})^k. \tag{87}$$

Since the deviation of the trajectory from $\mathcal{A}_k$ is already of order $R\epsilon$, it suffices to set the discretization resolution at the same scale, *i.e.,* $\epsilon' = \Theta(R\epsilon)$. Substituting this choice yields

$$|V| \leq (O(\frac{1}{\epsilon}))^k. \tag{88}$$

This directly leads to a discretization size whose exponent is $O(\text{sr}(D_T)/\epsilon^2)$, completing the argument in Lemma 5.

$\square$

## C EXPERIMENT DETAILS

### C.1 DATASETS

**CIFAR-10/100** (Krizhevsky & Hinton, 2009). Both datasets contain 50,000 images for training and 10,000 images for evaluation. They are used for natural images classification with categories of 10 and 100 respectively. The resolution of them is $32 \times 32$.

**ImageNet-1K** (Russakovsky et al., 2015). It contains 1,281,167 images for training and 50,000 images for validation and 10,000 for testing. It is the subset of the ImageNet-21K dataset with 1,000 categories.

**Alpaca** (Taori et al., 2023). It is a text dataset consisting of 52k instruction data generated by OpenAI's *text-davinci-003* engine, for the task of instruction fine-tuning large language model. To further reduce training costs, we conduct dataset quantization (Zhou et al., 2023) and select a subset of 1k instructions following (Qin et al., 2024).

**LESS** (Xia et al., 2024). It is an instruction-tuning dataset consisting of 270k data points collected from four datasets to reflect the heterogeneity in format, sequence length, and underlying tasks. We use it for the task of instruction fine-tuning large language model.

### C.2 DETAILS OF EXPERIMENT SETTINGS

**Implementation.** We implement our proposed SRS-Sampling approach as a *sampler* class in *PyTorch* API (Paszke et al., 2019), so that one can use it just like the most common random sampler. It is completely compatible with modern distributed training libraries, for example, *Accelerate* (Gugger et al., 2022). With a few lines of code, one can easily integrate our SRS-Sampling approach into the training pipeline like *Transformers* (Wolf et al., 2020).

**Hardware.** We conduct all the experiments on $8 \times$ *NVIDIA A100-SXM4-80GB* server, equipping with 128 2.6GHz Intel CPUs and 1TB main memory. We use bfloat16 mix-precision for training.

**Experiment settings.** We train CIFAR-10/100 with batch size of 1024 for 200 epochs, For CIFAR-10 with ResNet-18, we use the AdamW optimizer (Loshchilov & Hutter, 2019) with learning rate of 1e-4 and weight decay of 5e-5. For ResNet-50, we set both the weight decay and learning rate of AdamW to 1e-4. We train CIFAR-100 with learning rate of 5e-4 and weight decay of 1e-2 with AdamW for ResNet-18/50. In above case, we set beta1 to 0.9 and beta2 to 0.999 in AdamW optimizer. SGD optimizer with learning rate of 1e-3, momentum of 0.9, weight decay 5e-4 can reproduce the similar results. The lable smoothing factor is set to be 1e-2.

For ImageNet-1K, we set a large batch size of 4096 for 300 epochs and warm up for 20 epochs. we use accumulate step of 4 for acceptable memory consumption in GPU. For ResNet-50, ViT-Base(MAE) (He et al., 2022) and Swin-Tiny Transformer (Liu et al., 2021), we use AdamW optimizer with learning rate of 1e-4 and weight decay of 0.3 following (He et al., 2022), meanwhile SGD with learning rate 0.5, momentum 0.9, weight decay 2e-5 and batch size 1024 can reproduce the similar results.

We implement the image training pipeline partially based on timm and Huggingface. All images are transformed with commonly adopted data augmentations like random resized crop, random horizontal flip, together with RandAugment (Cubuk et al., 2019) and RandomErasing (Zhong et al., 2017) to avoid overfitting. For ImageNet-1K, we additionally adopt MixUp, CutMix (Zhang et al., 2018; Yun et al., 2019) for better generalization, together with exponential moving average (EMA) with max decay value of 0.9999.

For Alpaca, we fine-tune Llama-7B (Touvron et al., 2023). It is trained for 64 batch size and 15 epochs, we set AdamW with learning rate of 1.5e-4 and weight decay of 1e-2. We perform update through LoRA (Hu et al., 2022) with rank of 16, alpha 16, dropout 0.05.

For LESS, we fine-tune Mistral-7B (Jiang et al., 2023). It is trained with a batch size of 128 for 15 epochs, we use AdamW with a learning rate of 2e-5 and a weight decay of 1e-2. We perform updates via LoRA (Hu et al., 2022) with a rank of 128, alpha 512, and dropout 0.1.

For all experiments, we use Cosine schedule with warm up period of 5% of total iterations. We set the default value of $MR + LR^2$ in our proposed SRS-Sampling approach to be 1.0 if not specified, and subsets are resampled at the beginning of each epoch for all online methods, meanwhile total training iterations across methods will be aligned.

**Compared approaches.** The implementation of baselines are partially based on the DeepCore library (Guo et al., 2022). We set Random* to be the online version of Random, which conducts random selection in each epoch. Note that many existing coreset approaches for deep learning are based on greedy selection, including CD (Agarwal et al., 2020), Herding (Welling, 2009), K-Center (Sener & Savarese, 2018). Due to the heavy computational cost of greedy algorithm, they are typically performed offline. Meanwhile Craig (Mirzasoleiman et al., 2020), Glister (Killamsetty et al., 2021b) and $\epsilon$-greedy (Raju et al., 2021) perform online data selection. Least Confidence (Coleman et al., 2020), Margin (Coleman et al., 2020), Forgetting (Toneva et al., 2019a), DeepFool (Ducoffe & Precioso, 2018), UCB (Raju et al., 2021) are uncertainty-based methods. We also compare our method with influence-based methods like Influence (Koh & Liang, 2017), DP (Yang et al., 2023a), GREATS (Wang et al., 2024) and score-based methods like GraNd, EL2N (Paul et al., 2021). Among all, InfoBatch (Qin et al., 2024) is a loss-based weighted sampling approach, which is the most similar method to ours. It can be taken as a special case by fixing the number of different loss regions to be "2".

## C.3 COMPLETE RESULTS OF CIFAR-10/100 COMPARISONS

In Table 7, we show the complete results of comparison with SOTA methods on CIFAR-10, CIFAR-100 with ResNet-18 and ResNet-50. Part of the reported results are quoted from original papers.

## C.4 IMAGE CLASSIFICATION FINE-TUNING

Note that the results of image classification pre-training task shown in Tables 3 and 7 validate the effectiveness and generalization ability of our proposed SRS-Sampling approach, we want to emphasize that: the bounded approximation error of proposed SRS-Sampling is the reason why it can greatly outperform Random* method. Thus, we further verify the performance in image classifi-

Table 7: The Accuracy (%) comparison with state-of-the-art methods on ResNet-18. Since our proposed SRS-Sampling has a dynamic size, we align all methods to have the same iterations counts. Random$^*$ denotes online random sampling.

| | Dataset | CIFAR10 | | | CIFAR100 | | |
|---|---|---|---|---|---|---|---|
| | Subset Ratio % | 30 | 50 | 70 | 30 | 50 | 70 |
| Offline | Random | $90.2_{\downarrow 5.4}$ | $93.3_{\downarrow 2.3}$ | $94.6_{\downarrow 1.0}$ | $69.7_{\downarrow 8.5}$ | $72.1_{\downarrow 6.1}$ | $73.8_{\downarrow 4.4}$ |
| | CD (Agarwal et al., 2020) | $90.8_{\downarrow 4.8}$ | $94.3_{\downarrow 1.3}$ | $95.0_{\downarrow 0.6}$ | $70.3_{\downarrow 7.9}$ | $72.3_{\downarrow 5.9}$ | $74.2_{\downarrow 4.0}$ |
| | Herding (Welling, 2009) | $80.1_{\downarrow 15.5}$ | $88.0_{\downarrow 7.6}$ | $92.2_{\downarrow 3.4}$ | $69.6_{\downarrow 8.0}$ | $71.8_{\downarrow 6.4}$ | $73.1_{\downarrow 5.1}$ |
| | K-Center (Sener & Savarese, 2018) | $90.9_{\downarrow 4.7}$ | $93.9_{\downarrow 1.7}$ | $94.7_{\downarrow 0.9}$ | $70.2_{\downarrow 8.0}$ | $72.2_{\downarrow 6.0}$ | $74.1_{\downarrow 4.1}$ |
| | Least Confidence (Coleman et al., 2020) | $90.3_{\downarrow 5.3}$ | $94.5_{\downarrow 1.1}$ | $95.0_{\downarrow 0.6}$ | $69.8_{\downarrow 8.4}$ | $72.3_{\downarrow 5.9}$ | $74.2_{\downarrow 4.0}$ |
| | Margin (Coleman et al., 2020) | $90.9_{\downarrow 4.7}$ | $94.3_{\downarrow 1.3}$ | $94.9_{\downarrow 0.7}$ | $70.2_{\downarrow 8.0}$ | $72.2_{\downarrow 6.0}$ | $74.0_{\downarrow 4.2}$ |
| | Forgetting (Toneva et al., 2019a) | $91.7_{\downarrow 3.9}$ | $94.1_{\downarrow 1.5}$ | $94.7_{\downarrow 0.9}$ | $69.9_{\downarrow 8.3}$ | $73.1_{\downarrow 5.1}$ | $75.3_{\downarrow 2.9}$ |
| | DeepFool (Ducoffe & Precioso, 2018) | $90.0_{\downarrow 5.6}$ | $94.1_{\downarrow 1.5}$ | $95.1_{\downarrow 0.5}$ | $69.8_{\downarrow 6.4}$ | $73.2_{\downarrow 5.0}$ | $74.2_{\downarrow 4.0}$ |
| | Influence (Koh & Liang, 2017) | $88.3_{\downarrow 7.3}$ | $91.3_{\downarrow 4.3}$ | $93.1_{\downarrow 2.5}$ | $68.9_{\downarrow 9.5}$ | $72.0_{\downarrow 6.2}$ | $74.4_{\downarrow 3.8}$ |
| | DP (Yang et al., 2023a) | $90.8_{\downarrow 4.8}$ | $93.8_{\downarrow 1.8}$ | $94.9_{\downarrow 0.7}$ | - | $73.1_{\downarrow 5.1}$ | $77.2_{\downarrow 1.0}$ |
| | GraNd-4 (Paul et al., 2021) | $91.2_{\downarrow 4.4}$ | $94.6_{\downarrow 1.0}$ | $95.3_{\downarrow 0.3}$ | $68.8_{\downarrow 9.4}$ | $71.4_{\downarrow 6.8}$ | $74.6_{\downarrow 3.6}$ |
| | EL2N-2 (Paul et al., 2021) | $89.8_{\downarrow 5.8}$ | $93.2_{\downarrow 2.4}$ | $94.4_{\downarrow 1.2}$ | $68.5_{\downarrow 9.7}$ | $71.0_{\downarrow 7.2}$ | $74.1_{\downarrow 4.1}$ |
| | EL2N-20 (Paul et al., 2021) | $91.9_{\downarrow 3.7}$ | $\mathbf{95.1}_{\downarrow 0.5}$ | $95.3_{\downarrow 0.3}$ | - | $72.1_{\downarrow 6.1}$ | $77.2_{\downarrow 1.0}$ |
| Online | Random$^*$ | $92.6_{\downarrow 3.0}$ | $93.7_{\downarrow 1.9}$ | $94.7_{\downarrow 0.9}$ | $73.2_{\downarrow 5.0}$ | $75.3_{\downarrow 2.9}$ | $77.3_{\downarrow 0.9}$ |
| | Craig (Mirzasoleiman et al., 2020) | $88.4_{\downarrow 7.2}$ | $93.3_{\downarrow 3.3}$ | $94.8_{\downarrow 0.8}$ | $69.7_{\downarrow 8.5}$ | $71.9_{\downarrow 6.3}$ | $74.4_{\downarrow 3.8}$ |
| | Glister (Killamsetty et al., 2021b) | $90.9_{\downarrow 4.7}$ | $94.0_{\downarrow 1.6}$ | $95.2_{\downarrow 0.4}$ | $70.4_{\downarrow 7.8}$ | $73.2_{\downarrow 5.0}$ | $74.6_{\downarrow 3.6}$ |
| | $\epsilon$-greedy (Raju et al., 2021) | $94.1_{\downarrow 1.5}$ | $94.9_{\downarrow 0.7}$ | $95.2_{\downarrow 0.4}$ | - | $74.8_{\downarrow 3.4}$ | $76.4_{\downarrow 1.8}$ |
| | UCB (Raju et al., 2021) | $93.9_{\downarrow 1.7}$ | $94.7_{\downarrow 0.9}$ | $95.3_{\downarrow 0.3}$ | - | $75.3_{\downarrow 2.9}$ | $77.3_{\downarrow 0.9}$ |
| | InfoBatch (Qin et al., 2024) | $94.3_{\downarrow 1.3}$ | $94.9_{\downarrow 0.7}$ | $\mathbf{95.6}_{\uparrow 0.0}$ | $74.6_{\downarrow 3.6}$ | $76.9_{\downarrow 1.3}$ | $77.9_{\downarrow 0.3}$ |
| | SRS-Sampling | $\mathbf{94.6}_{\downarrow 1.0}$ | $\mathbf{95.1}_{\downarrow 0.5}$ | $\mathbf{95.6}_{\uparrow 0.0}$ | $\mathbf{75.7}_{\downarrow 2.5}$ | $\mathbf{77.6}_{\downarrow 0.6}$ | $\mathbf{78.2}_{\uparrow 0.0}$ |
| | Overall Dataset | $95.6_{\pm 0.1}$ | | | $78.2_{\pm 0.1}$ | | |

cation fine-tuning task. Intuitively, fine-tuning scenario with limited training budget should have a strict requirement for loss approximation with bounded error, in order to preserve and generalize the ability of the pre-trained model to the fine-tuning dataset to the most extent.

We use CIFAR-10 and CIFAR-100 dataset for experiments and subset size is set to 60%. We use ResNet-50 pre-trained on ImageNet-1K (He et al., 2016), then fine-tune CIFAR-10 for 10 epochs and CIFAR-100 for 50 epochs on respective dataset. Other training details are same with Table 7. The results of our proposed SRS-Sampling approach and Random$^*$ are reported in Table 8, the consistent and significant improvement against Random$^*$ further support the importance of bounded approximation error of our proposed SRS-Sampling approach.

Table 8: Fine-tuning CIFAR-10 for 10 epochs and CIFAR-100 for 50 epochs, subset size is set to 60%. We use ResNet-50 pre-trained on ImageNet-1K, and Accuracy (%) is reported through the fine-tuning progress (%). Random$^*$ denotes online random sampling.

| Dataset | Method | Fine-tuning Progress (%) | | | | |
|---|---|---|---|---|---|---|
| | | 20 | 40 | 60 | 80 | 100 |
| CIFAR-10 | Random$^*$ | 67.8 | 77.6 | 93.0 | 94.3 | 94.9 |
| | SRS-Sampling (ours) | $76.8_{\uparrow 9.0}$ | $90.1_{\uparrow 12.5}$ | $93.5_{\uparrow 0.5}$ | $94.5_{\uparrow 0.2}$ | $95.1_{\uparrow 0.2}$ |
| CIFAR-100 | Random$^*$ | 82.3 | 83.7 | 85.5 | 86.3 | 86.3 |
| | SRS-Sampling (ours) | $82.8_{\uparrow 0.5}$ | $84.4_{\uparrow 0.7}$ | $85.7_{\uparrow 0.2}$ | $86.6_{\uparrow 0.3}$ | $86.8_{\uparrow 0.5}$ |

## C.5 INSTRUCTION FINE-TUNING

Besides image classification task, we validate the effectiveness of our SRS-Sampling approach in instruction fine-tuning task. As mentioned before, the Alpaca dataset is generated by OpenAI's model. It contains the instruction task like generate story, rewrite sentence and explain concept. The detail of dataset can be found in (Taori et al., 2023). We fine-tune Llama-7B with LoRA adapter

on the 2% subset of Alpaca after dataset quantization (DQ) (Zhou et al., 2023). We set the baseline to be online random sampling (DQ+Random*), and also compare our SRS-Sampling method (DQ+SRS) with InfoBatch (DQ+InfoBatch), GREATS (DQ+GREATS). Since GREATS requires additional validation data to compute the influence, we randomly sample from training set as its validation data for fairness, and the number of validation data is set to be 16 as default in Wang et al. (2024). And the training iterations across three methods are aligned to 80% of total iterations of regular training. We show the results in Table 4. BBH (version 3.0) and DROP (version 3.0) use Accuracy to assess the reasoning ability, MMLU (version 2.0) use Accuracy to assess the knowledge for Multiple Choice Question, Human-Eval (version 1.0) use Pass@1 to assess the coding ability. All these metrics are the higher the better. The average scores from tested benchmark show that, our proposed SRS-Sampling method is more effective and competitive than online random selection and InfoBatch, together with a negligible runtime overhead in exchange for better training performance.

We also fine-tune Mistral-7B with LoRA adapter on the LESS dataset. The subset ratio is set to 1%, which is extremely small. We report the scores on BBH (version 3.0), HellaSwag (version 1.0), MMLU (version 2.0) and Human-Eval (version 1.0). All these metrics are higher-is-better. The average scores from the tested benchmark in Table 5 show that our proposed SRS-Sampling method has a notable advantage over Random*, especially when the subset ratio is extremely small.

## C.6 ABLATION AND ROBUSTNESS

We perform ablation study to validate that training a model benefits from **dynamic subset size allocation** related to stable rank in Table 9, where general improvement up to 0.5% exists across ResNet-18 and ResNet-50. The **robustness** of hyper-parameter in our SRS-Sampling approach is also reported in Table 10. Recall that $M$ is the maximal gradient norm, and $R$ is the radius in parameter space. We study $M$ and $R$ together, since according to the formulation of $|Q_j|$, $MR + LR^2$ determines the smoothness of sampling ratio across different loss regions. The results shows that it is robust in a broad range from 0.5 to 5.0, which is the range of loss value typically.

Table 9: Ablation of dynamic subset size allocation through training: whether to compute the stable rank of trajectory. We align the SGD iterations across the ablation to be 30% of full training.

| Ablation | | Accuracy | |
|---|---|---|---|
| Dataset | Dyn. Alloc. | ResNet-18 | ResNet-50 |
| CIFAR-10 | w/o | 94.1±0.3 | 94.6±0.2 |
| CIFAR-10 | w/ | 94.6±0.2 | 94.8±0.2 |
| CIFAR-100 | w/o | 75.3±0.3 | 78.0±0.2 |
| CIFAR-100 | w/ | 75.7±0.2 | 78.5±0.2 |

Table 10: Robustness of hyperameter in SRS-Sampling. Experiments are conducted on CIFAR-10 with 30% subset size.

| Robustness | Accuracy | |
|---|---|---|
| $MR + LR^2$ | ResNet-18 | ResNet-50 |
| 0.1 | 94.3±0.3 | 94.6±0.2 |
| 0.5 | 94.5±0.2 | 94.7±0.2 |
| 1.0 | 94.6±0.2 | 94.8±0.2 |
| 5.0 | 94.5±0.2 | 94.8±0.3 |
| 10.0 | 94.4±0.3 | 94.7±0.2 |

We also report the performance of SRS-Sampling on noisy datasets in Table 11. We conduct experiments on CIFAR-10 with different ratios of label noise, where the subset ratio is set to 70% and we train ResNet-50 for 200 epochs. The accuracy (%) is reported over three runs.

Table 11: Test Accuracy (%) of ResNet-50 trained for 200 epochs on CIFAR-10 with label noise. Results are averaged over three runs. Random* denotes online random sampling.

| Corruption Ratio | 0.1 | 0.2 | 0.3 | 0.4 | 0.5 |
|---|---|---|---|---|---|
| Random* | $93.8_{\pm0.3}$ | $88.6_{\pm0.3}$ | $81.3_{\pm0.3}$ | $77.0_{\pm0.3}$ | $69.9_{\pm0.3}$ |
| SRS-Sampling | $94.6_{\pm0.3\ \uparrow0.8}$ | $89.5_{\pm0.3\ \uparrow0.9}$ | $83.8_{\pm0.3\ \uparrow2.5}$ | $80.3_{\pm0.3\ \uparrow3.3}$ | $72.9_{\pm0.3\ \uparrow3.0}$ |

The results in Table 11 suggest that vanilla uniform sampling (Random*) is more vulnerable to noise compared to SRS-Sampling, which provides approximation error guarantees. SRS-Sampling enables the training of neural networks more robust to the label noise, and the advantage on test accuracy is more notable when the corruption ratio is higher.

## C.7 SRS-SAMPLING DETAILS AND STABLE RANK

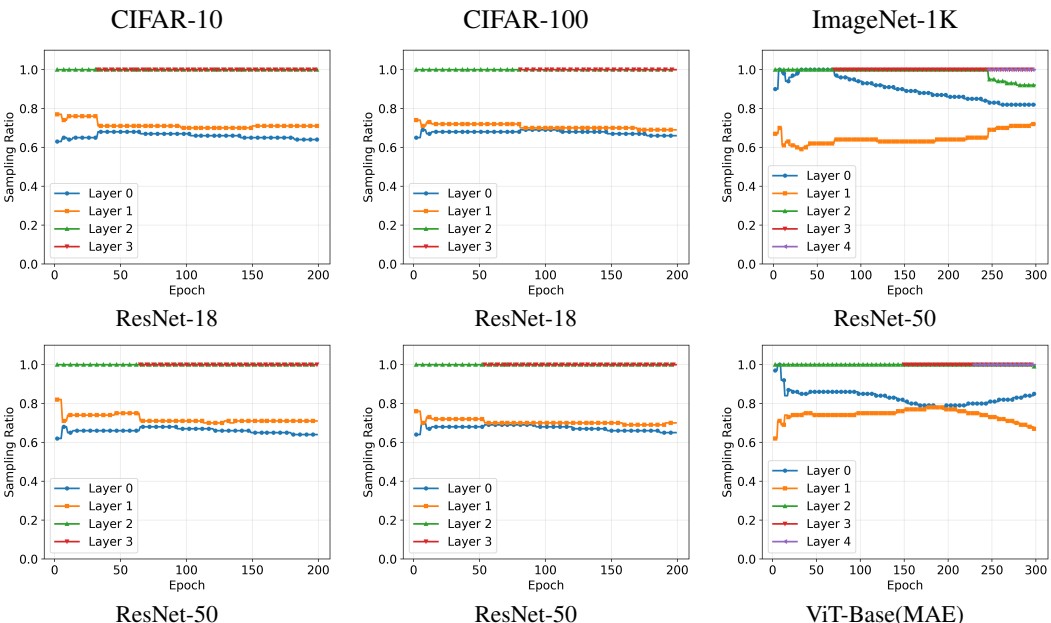

Figure 4: Sampling probability in different loss regions across training.

To illustrate the details of sampling strategy of our proposed SRS-Sampling approach, we visualize the sampling probability in different loss regions $P_j$ across training. For clarity, we ignore the lines corresponding to the regions where $|P_j| = 0$. The results for CIFAR-10, CIFAR-100 with 70% subset ratio, ImageNet-1K with 80% subset ratio are shown in Fig. 4, where $Layer\ j$ corresponds to $P_j$. If the number of layers is only "2", it corresponds to the case of InfoBatch (Qin et al., 2024).

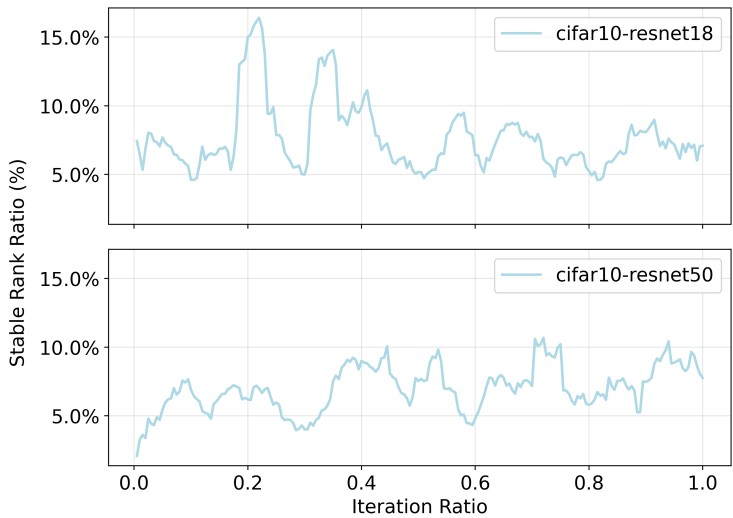

Figure 5: Stable rank ratio during training on CIFAR-10 with ResNet-18 and ResNet-50.

We also visualize the stable rank ratio (stable rank compared with parameter dimension) of gradient trajectory across training on CIFAR-10 in Fig. 5, the result is of independent interest for better understanding the training dynamics. We should remind the readers that, when training a deep learning model, it may not necessarily converge to even the local minimum at the end of a training schedule, due to techniques like early stop to avoid overfitting. Furthermore, the stable rank of gradient trajectory does not intent to measure the convergence of learning. Since even near a local

minimum in parameter space, the objective function can still be complicated, and the stable rank of gradient trajectory may not necessarily be small. From the visualization Fig. 5, one can notice that the stable rank ratio fluctuates around 8% of the total parameter size in CIFAR-10. Stable rank ratio on ResNet-18 can sometimes higher up to 15% in the early stage of training, it may due to the reason that ResNet-18 is smaller than ResNet-50, and have a relatively small ratio of redundant dimensions. As visualized in Fig. 5, the stable rank upper bound across the early and late stages of training remains consistently low compared to the parameter size. Based on the dynamics of stable rank, one can allocate an appropriate subset size fluctuating around a predefined value, which depends on the evolving dimension of training subspace. However, we did not observe a clear monotonically increasing or decreasing trend in the experiment. In the future, we think an interesting direction is to investigate how training complexity evolves across different training stages, through a fine-grained stable rank analysis. In particular, we need to develop a new theoretical framework to characterize the evolving dynamics of stable rank throughout the training process.

