# OpenReview forum: "Bridging Between Stable Rank and Data Selection: A Novel Sampling Method for Fast Training of Deep Neural Networks"
_ICLR.cc/2026/Conference — Submitted to ICLR 2026_

### Official Review · Reviewer_6krn · 2025-10-31

**Soundness:** 3
**Presentation:** 3
**Contribution:** 3
**Rating:** 4
**Confidence:** 3

**Summary:**

This work proposes stable rank related stratified sampling (SRS-Sampling), a loss-stratified data selection method. This approach ensures that the required subset size is proportional to the stability rank of the gradient trajectory (GT) matrix, rather than the parameter dimension. SRS-Sampling was proposed based on a clear theoretical foundation, and methods for efficient computation were also presented.

**Strengths:**

1. This manuscript is well written with clear theoretical evidences. Also, by clearly outlining the limitations of prior works and the challenges encountered in narrowing the scope to detour these limitations, it is easy to follow the motivation of this research.

2. The proposed idea that utilize the tracking dimensionality of gradient trajectory subspace for data selection, which is unaffected by the number of parameters, is novel and interesting.

**Weaknesses:**

1. The performance of the SRS-sampling does not differ significantly from the baselines. Most performance gaps are less than 0.5%p, and it is particularly similar to the baseline on the ImageNet-1K and Alpaca datasets. To claim statistical significance for SRS-Sampling, additional p-value-based analysis is required.

2. The explanation of loss-based exponential partitioning is insufficient. It is not readily apparent how this partitioning aids in approximating the objective function of the entire dataset using a weighted subset, leading it to be perceived as a heuristic approach. Furthermore, with large-scale datasets, the number of partitions increases simultaneously, and certain partitions end up with very large intervals. This leads to the presumption that numerous partitions will be nearly empty, resulting in high redundancy.

**Questions:**

1. I would like to know whether SRS-Sampling remains effective even in situations where the subset ratio is extremely small such as 1% and 5%.

2. I believe a sensitivity analysis is required based on the value of the base used in the exponential partition.

3. To obtain an upper bound for the stable rank, the Jacobian, hidden-layer representations, and gradients are required. Storing and computing these components is expected to necessitate greater memory usage than the baseline. Therefore, a comparative analysis of computational cost is required.

---

> ### Author Response · Authors · 2025-11-23
> **Response to Reviewer 6krn (1/4)**
>
> We thank the reviewer for the constructive feedback. Below, we try to address your concerns.
>
> **W1 [Additional p-value-based analysis is required]**
>
> **A.**
>
> 1. Actually, SRS-Sampling **accelerates convergence significantly**, as shown in our experimental results. We report the results for CIFAR-10 and CIFAR-100 fine-tuning with ResNet-50 pre-trained on ImageNet-1K in Table 8. The results in Table 8 suggest that SRS-Sampling provides a significant advantage over Random$ ^ *$ in the early stage of training, achieving up to **12.5%p improvement** at 40% of the training process. This indicates the effectiveness of SRS-Sampling in accelerating convergence, particularly when early-stage gradients may be too chaotic for vanilla uniform sampling.
> 2. Below, we report the detailed standard deviation error analysis together with **p-values from paired t-tests** across three independent runs for 200 epochs, where other setting is the same as the ImageNet-1K experiment in Table 3.
>
> **Performance Results (Mean ± Std)**
>
> Table R8: ImageNet-1K results of standard deviation error analysis from three independent runs. Three models are trained from scratch for 200 epochs, ViT-B is additionally fine-tuned for 50 epochs. Random$ ^ *$ denotes online random sampling.
>
> | Model     | Subset Ratio | Random*       | InfoBatch     | SRS-Sampling      |
> | --------- | ------------ | ------------- | ------------- | ----------------- |
> | ResNet-50 | 60.0%        | 72.48 ± 0.438 | 74.86 ± 0.437 | **76.56 ± 0.252** |
> | Swin-T    | 80.0%        | 75.71 ± 0.435 | 78.14 ± 0.385 | **80.84 ± 0.311** |
> | ViT-B     | 80.0%        | 79.04 ± 0.454 | 80.78 ± 0.420 | **82.36 ± 0.245** |
>
> **Statistical Significance (p-values) **
>
> Table R9: ImageNet-1K results of p-values from paired t-tests from three independent runs. Three models are trained from scratch for 200 epochs, ViT-B is additionally fine-tuned for 50 epochs. Random$ ^ *$ denotes online random sampling.
>
> | Model     | SRS-Sampling vs Random* | SRS-Sampling vs InfoBatch |
> | --------- | ----------------------- | ------------------------- |
> | ResNet-50 | 0.0093 (<0.01)          | 0.0482 (<0.05)            |
> | Swin-T    | 0.0059 (<0.01)          | 0.0004 (<0.001)           |
> | ViT-B     | 0.0128 (<0.05)          | 0.0273 (<0.05)            |
>
> Our paired t-test analysis suggests that SRS-Sampling achieves statistically significant improvements over both Random$ ^ {*}$ and InfoBatch baselines across all three models (ResNet-50, Swin-T, and ViT-B) in large-scale ImageNet-1K training, with all p-values below $0.05$, which validates the **statistical significance**.

---

> ### Author Response · Authors · 2025-11-23
> **Response to Reviewer 6krn (2/4)**
>
> **W2 [Further explanation of exponential partitioning with theoretical guarantee]**
>
> **A.** Thank you for this good question.
>
> 1. **[Theoretical guarantee]** The exponential partition is not heuristic; it is related to the bounded loss approximation error guarantee provided by SRS-Sampling. **First**, it directly enables the Hoeffding-based concentration bound for each loss stratum, as detailed in the proof of Lemma 2. The upper bound of loss value on each stratum is at most $\mathtt{Base}$ times of the lower bound, where $\mathtt{Base}=2$ by default in our experiments. It ensures that the sample size $|Q _ j|$ remains relatively small via Hoeffding's bound, as shown in Section 4.1 and in the complete proof of Theorem 1. **Second**, the exponential partition ensures a bounded number of partitions $N = \lceil \log _ 2 n \rceil$, as shown in Line 205, where this logarithmic bound can be regarded as a relatively small number.
> 2. **[Address the concerns about partitions redundancy]** For large-scale datasets, we have at most $N = \lceil \log _ 2 n \rceil$ non-empty partitions. Although exponential partition ensures large intervals for strata with large loss, there may still be nearly empty partitions. However, it does not incur any significant computational or storage redundancy. According to the exponential partitioning defined in eq (3) and eq (4) (Page 4), Algorithm 1 allocates each sample $(x_i, y_i)$ with loss value $f _ i(\theta _ {\mathtt{ini}})$  to stratum $Q _ j$ where $j=\lceil \log _ {2} \frac{f _ i(\theta _ {\mathtt{ini}})}{F(\theta _ {\mathtt{ini}})} \rceil$. This assignment can be done efficiently in one pass through the dataset regardless of the partition number $N$.
>
>
>
> **Q1 [Performance when subset ratio is extremely small]**
>
> **A.** Thank you for this interesting question. **First**, actually we do have experiments related to the extremely small subset setting. For Alpaca dataset, dataset quantization (DQ) is employed to curate a small (2%) dataset from the original Alpaca dataset first (Section C.5), and then we align iterations across compared methods to 80% of regular training. Ultimately the equivalent subset size is 1.6%, which is extremely small. The results in **Table 4** suggest that SRS-Sampling is compatible with the dataset quantization method (DQ), and in general achieves better performance than all other compared baselines.
>
> **Second**, to further investigate the performance under this setting, we fine-tune *Mistral-7B* [1] (updated in July 2025) on the *LESS* dataset [2] (released in 2024), which is a combination of four instruction tuning datasets and comprises $270k$ data points. We set the subset ratio to **1%**, so as to examine the effectiveness of SRS-Sampling with an extremely small subset ratio, which is a potential application scenario, particularly in large-scale LLM fine-tuning. Benchmark results (Accuracy %) for SRS-Sampling and Random$ ^ *$ are reported below, all other fine-tuning settings follow those in [2].
>
> Table R10: Fine-tuning Mistral-7B on LESS for 15 epochs. Subset ratio is set to 1%. Random$ ^ *$ denotes online random sampling. Results of total node time for fine-tuning and Zero-shot Accuracy (%) on four benchmarks are reported.
>
> | Method       | Time(h) | BBH      | HellaSwag | MMLU-Overall | MMLU-Hum. | MMLU-Other. | MMLU-Soc. | MMLU-STEM | Human-Eval | Avg.     |
> | ------------ | ------- | -------- | --------- | ------------ | --------- | ----------- | --------- | --------- | ---------- | -------- |
> | Random$ ^ *$ | 10.31   | 53.4     | 52.3      | 60.4         | 52.6      | 67.5        | 68.4      | 53.2      | 64.3       | 57.6     |
> | SRS-Sampling | 10.32   | **55.8** | **53.2**  | **61.7**     | **54.9**  | **68.6**    | **69.7**  | **53.7**  | **65.4**   | **59.0** |
>
> The benchmark results show that SRS-Sampling generalizes well to different LLM architectures and newly released fine-tuning datasets. Moreover, under extremely small subset ratio scenarios, SRS-Sampling exhibits a more significant advantage over Random$ ^ *$ (vanilla uniform sampling), with improvements of up to **1.4%p** in benchmark scores.
>
> [1] Jiang, et al. “Mistral 7B.” *ArXiv* abs/2310.06825, 2023.
>
> [2] Xia, Mengzhou, et al. "LESS: selecting influential data for targeted instruction tuning." ICML, 2024.

---

> ### Author Response · Authors · 2025-11-23
> **Response to Reviewer 6krn (3/4)**
>
> **Q2 [Sensitivity analysis for the base of exponential partition]**
>
> **A.** The number of loss strata depends on the base of the exponential partition. Here, we perform the sensitivity analysis with respect to the base, using the CIFAR-10 dataset and the ResNet-50 model with three independent runs, where the dataset size is $n=50000$ and the number of loss strata is $N+1=\lceil \log _ {\mathtt{Base}} n \rceil+1$. The subset ratio is set to $10\%$, and the Random$ ^ {*}$ baseline is also included for comparison.
>
>
>  As validated by our experiments, the performance is relatively robust on the setting of $\mathtt{Base}$, as long as it is not set so large that the number of partitions is too small. Typically, we can set $\mathtt{Base} \in (1,5]$. For smaller datasets, a smaller $\mathtt{Base}$ is preferred to obtain finer partitions.
>
>
> Table R11: Sensitivity analysis about the base of exponential partitions on CIFAR-10. ResNet-50 is trained for 200 epochs with three independent runs. Random$ ^ *$ denotes online random sampling. Test Accuracy (%) is reported.
>
> | Base         | 1.4                | 1.6                | 1.8                | 2                  | 5                  | 10                 | Random$ ^ {*}$     |
> | ------------ | ------------------ | ------------------ | ------------------ | ------------------ | ------------------ | ------------------ | ------------------ |
> | #Partitions  | 34                 | 25                 | 20                 | 17                 | 8                  | 6                  | 1                  |
> | Accuracy (%) | $92.7 _ {\pm 0.3}$ | $93.6 _ {\pm 0.3}$ | $93.0 _ {\pm 0.3}$ | $93.4 _ {\pm 0.3}$ | $92.8 _ {\pm 0.3}$ | $91.9 _ {\pm 0.3}$ | $85.6 _ {\pm 0.3}$ |

---

> ### Author Response · Authors · 2025-11-23
> **Response to Reviewer 6krn (4/4)**
>
> **Q3.1 [Computational and storage complexity of Algorithm 2]**
>
> **A.** Thank you for this good question.
>
> 1. **[Computational complexity]** For the estimation of an upper bound on the stable rank, Algorithm 2 runs in $\mathtt{Time} _ {sr}=O(d ^ {(l)}sk)$, which uses the Lanczos algorithm [1] to compute eigen-pairs. $d ^ {(l)}$ is the row size of the Kronecker product $S= C \otimes B$ ($\textit{i.e.,}$ the dimension of gradient vector of layer $l$), $s \leq d ^ {(l)}$ is the average number of nonzero elements in a row of $S$, and the first $k\ll d ^ {(l)}$ eigen-pairs are computed (note that $1-\eta \lambda _ 1, 1-\eta \lambda _ 2$ are the two largest distinct eigenvalues of $I-\eta S$ as shown in the proof of Theorem 2. Therefore, we have $k=2$ iterations typically). By utilizing the basic property about eigen-pairs of Kronecker product (Theorem 4.2.12 of [2]), the problem size, i.e., $d ^ {(l)}$ of $S$ can be further decreased to the column size and row size of parameter matrix at layer $l$ (roughly speaking, the hidden size $h$ of $C$ and $B$). Therefore, the computational complexity can be further reduced to $O(k * h ^ 2)$. Typically we have $h ^ 2 = O(d/L)$ for an $L$ layers neural network with total parameter size of $d$. **Overall**, even if we disregard the parallel-processing capabilities of GPUs for matrix–vector multiplication, and sequentially compute for $L$ layers, the time complexity for Algorithm 2 is only $\mathtt{Time} _ {sr}=O(k * L*d/L)=O(kd)$. If we consider the parallel computation based on GPU, the computational complexity can be further reduced.
> 2. **[Storage complexity]** During the regular training process of neural networks, the Jacobian and hidden-layer representations are already required to compute the per-layer gradients [3]. Specifically, a fixed memory buffer is allocated for their computation and storage as done in the regular back-propagation process. We simply reuse these values without additional computation or storage, and directly input them to Algorithm 2. Also note that the Lanczos algorithm [1] used in Algorithm 2 for eigen-pairs computation does not require the instantiation of matrices but only requires a matrix-vector product function. Even if the matrices were instantiated, this would require only $O(d)$ memory by utilizing the property of Kronecker product. Therefore, we compute the eigen-pairs on-the-fly without additional storage allocation.
>
> [1] Calvetti, Daniela, Lothar Reichel, and Danny Chris Sorensen. "An implicitly restarted Lanczos method for large symmetric eigenvalue problems." *Electronic Transactions on Numerical Analysis*, 1994.
>
> [2] Roger A Horn and Charles R Johnson. "Topics in matrix analysis." Cambridge university press, 1994.
>
> [3] Wang, et al. "Greats: Online selection of high-quality data for llm training in every iteration." NeurIPS, 2024.
>
>
>
> **Q3.2 [Comparative analysis of computational cost with baselines]**
>
> **A.** We provide the computational analysis across the baselines below. Note that the proposed SRS-Sampling has the computational complexity of $O(n+kd)$ (we take $k=2$ as a small constant typically). Among this baselines, InfoBatch has a computational complexity of $O(n)$.  K-Center, Least Confidence, Margin, CD, Craig, GREATS, Glister take $O(bnd)$ running time, for the selection of $b$ samples from a dataset of size $n$. Influence method takes $O(nd \kappa)$ running time, where $\kappa$ is the factor for inverse Hessian approximation. DP takes $O(nd \kappa + T _ {\mathtt{iter}})$ for additionally solving a constraint problem. UCB, $\epsilon-$greedy, EL2N, Forgetting and GraNd take $O(nd + b\log n)$ running time, including the cost of computing data scores and selecting the top-$b$ samples.
> **Overall**, apart from Random$^*$ which just performs simple uniform sampling, SRS-Sampling's $O(n+kd)$ complexity is only slightly higher than InfoBatch's $O(n)$ complexity (as mentioned above, we typically take $k=2$ for solving two distinct eigen-pairs, and the extra $O(kd)$ complexity can be further reduced by fast GPU implementation).

---

> ### Author Response · Authors · 2025-11-27
>
> Thank you for taking the time to review our paper and provide valuable feedback. As the discussion phase nears its conclusion, we would like to confirm that our responses have addressed your concerns. If you have any additional comments, we will do our best to address them.

---

### Official Review · Reviewer_Hi6J · 2025-11-01

**Soundness:** 3
**Presentation:** 3
**Contribution:** 3
**Rating:** 4
**Confidence:** 3

**Summary:**

The main idea of the paper is to develop a novel data selection method for efficient training of deep neural networks by leveraging the evolving dimensionality of the Gradient Trajectory (GT) subspace. The authors propose measuring the dimension of the training space using the stable rank of the gradient trajectory matrix, which represents a continuous and soft version of rank, reflecting the principal components of the gradient trajectory.

The paper proposes Stable Rank related Stratified Sampling (SRS-Sampling) method that selects a weighted subset of data for training, with the required sample size scaling linearly with this stable rank.

The empirical results on academic benchmarks such as MNIST, CIFAR, etc., show improvements. Experiments on Alpaca dataset with LLama-7B are encouraging.

**Strengths:**

1. The proposed method uses a smart stratified sampling approach based on how hard the network currently finds each data point (measured via loss), which ensures training speedups without losing accuracy, making it practical for large neural networks.

2. The empirical results show value of the proposed methods.

**Weaknesses:**

1. The connection between the gradient trajectory (GT) theory and the stratified sampling method is somewhat indirect. The paper leverages the stable rank of the GT matrix to characterize the dimension of the crucial training subspace, which guides the overall sample size needed. However, the stratified sampling based on loss partitions is not directly derived from the GT analysis but rather integrated as a practical sampling strategy inspired by importance sampling. This gap can make it hard to fully grasp how the GT stable rank determines the specific loss-based sampling probabilities and partitioning.

2. While the paper provides theoretical guarantees about sample size scaling with the stable rank and good empirical results, the computational overhead and complexity of estimating stable rank and eigen-pairs for very large models might limit practical real-time usage.

3. The adaptive and exponential partitioning of loss regions is a heuristic design choice without a fully explicit theoretical link to the GT dimension, so the choice of the number/count of regions and the partition base might affect results and requires tuning.

**Questions:**

1. The evaluation of LLM fine-tuning is conducted on older models and datasets; how well would the method generalize to more recent, larger models and evolving datasets?

2. How sensitive is the method’s efficiency and accuracy to the choice of the partitioning base and the number of loss regions in stratified sampling?

3. What is the computational overhead of estimating the stable rank in very large-scale settings, and how does it affect overall training speedups?

4. Would the approach degrade in scenarios with very noisy or imbalanced datasets where loss values may have less meaningful gradients?

5. Does the adaptive sample size based on stable rank sufficiently capture dynamic changes in training complexity during different training phases, e.g., early vs. late?

---

> ### Author Response · Authors · 2025-11-23
> **Response to Reviewer Hi6J (1/3)**
>
> We thank the reviewer for the constructive feedback. Below, we try to address your concerns.
>
> **W1 [Connection between GT analysis and stratified sampling]**
>
> **A.** Thank you for this good question. The proof for approximation error of the objective function deeply depends on the GT analysis. Let us explain their connection with more detail.
>
> For a sampling method aiming for loss approximation, there are two key aspects: unbiased estimator and bounded approximation error. Importance sampling is a fundamental sampling idea shared across different data selection methods, as mentioned in our introduction part. The proposed stratified sampling method also utilizes the importance sampling procedure, which ensures an unbiased estimator for the loss value of *current* parameter $\theta _ {\mathtt{ini}}$. However, the difficulty lies in how to quantify the required sample size to provide bounded approximation error for the **whole** *GT subspace* (instead of a single $\theta _ {\mathtt{ini}}$). In this paper, we establish the linear dependence between overall sample size and GT stable rank (Theorem 1), and show in Lemma 1 that the sample size $|Q _ j|$ in each loss stratum linearly depends on the stable rank. This relation between GT stable rank and $|Q _ j|$ further influences the sampling probability in each stratum (Algorithm 1).
>
>
>
>
>
> **W2 & Q3 [Computational overhead, real-time usage of Algorithm 2]** "What is the computational overhead of estimating the stable rank in very large-scale settings, and how does it affect overall training speedups?"
>
> **A.**
>
> 1. **[Computational overhead]** For stable rank upper bound estimation, Algorithm 2 runs in $\mathtt{Time} _ {sr}=O(d ^ {(l)}sk)$, which involves the Lanczos algorithm [1] for solving eigen-pairs. $d ^ {(l)}$ is the row size of the Kronecker product $S= C \otimes B$ ($\textit{i.e.,}$ the dimension of gradient vector of layer $l$), $s \leq d ^ {(l)}$ is the average number of nonzero elements in a row of $S$, and the first $k\ll d ^ {(l)}$ eigen-pairs are computed (note that $1-\eta \lambda _ 1, 1-\eta \lambda _ 2$ are the two largest distinct eigenvalues of $I-\eta S$ as shown in the proof of Theorem 2. Therefore, we typically have $k=2$ iterations). By utilizing the basic property about eigen-pairs of Kronecker product (Theorem 4.2.12 of [2]), the problem size, i.e., $d ^ {(l)}$ of $S$ can be further decreased to the column size and row size of parameter matrix at layer $l$ (roughly speaking, the hidden size $h$ of $C$ and $B$). Therefore, the computational complexity can be further reduced to $O(k * h ^ 2)$. Typically we have $h ^ 2 = O(d/L)$ for an $L$ layers neural network with total parameter size of $d$. **Overall**, even if we disregard the parallel-processing capabilities of GPUs for matrix–vector multiplication, and sequentially compute for $L$ layers, the time complexity for Algorithm 2 is only $\mathtt{Time} _ {sr}=O(k * L*d/L)=O(kd)$. If we consider the parallel computation based on GPU, the computational complexity can be further reduced.
> 2. **[Real-time usage]** The overall non-parallelized computational complexity of Algorithm 2 is $O(kd)$, and it can be further reduced by fast GPU implementation, where $d$ is the parameter count of a neural network, and $k$ is the number of iterations. Typically, we set $k=2$ to find $\lambda_1, \lambda_2$. It is lightweight for real-time usage as validated in Table 2, Table 4 and Table 5, where the results of training time already include the running time for weighted subset construction and stable rank estimation. Furthermore, the system-algorithm co-design could be incorporated to further optimize the time overhead. For example, the computation of Algorithm 2 can be offloaded to the CPU several iterations prior to the execution of Algorithm 1, thereby avoiding any blocking in the regular training pipeline.
>
> [1] Calvetti, Daniela, Lothar Reichel, and Danny Chris Sorensen. "An implicitly restarted Lanczos method for large symmetric eigenvalue problems." *Electronic Transactions on Numerical Analysis*, 1994.
>
> [2] Roger A Horn and Charles R Johnson. "Topics in matrix analysis." Cambridge university press, 1994.

---

> ### Author Response · Authors · 2025-11-23
> **Response to Reviewer Hi6J (2/3)**
>
> **W3 & Q2 [Sensitivity of partitioning base about efficiency and accuracy]**
>
> **A.**
>
> 1. **[Efficiency]** For a given dataset size $n$, the number of loss regions depends on the choice of the partitioning base, **i.e.**, $N = \lceil \log _ {\mathtt{Base}} n \rceil$. But we need to clarify that the choice of $\mathtt{Base}$ does not incur significant computational or storage redundancy. According to the exponential partitioning defined in eq (3) and eq (4) (Page 4), Algorithm 1 allocates each sample $(x_i, y_i)$ with loss value $f _ i(\theta _ {\mathtt{ini}})$  to stratum $Q _ j$ where $j=\lceil \log _ {\mathtt{Base}} \frac{f _ i(\theta _ {\mathtt{ini}})}{F(\theta _ {\mathtt{ini}})} \rceil$. This assignment can be done efficiently in one pass through the dataset regardless of the partition number $N$.
> 2. **[Accuracy]** We perform the sensitivity analysis with respect to the base, using the CIFAR-10 dataset and a ResNet-50 model with three independent runs, where the dataset size is $n=50000$ and the number of loss strata is $N+1=\lceil \log _ {\mathtt{Base}} n \rceil+1$, the subset ratio is set to $10\%$, and the baseline Random$ ^ {*}$ is also included for comparison. As validated by our experiments, the performance is relatively robust on the setting of $\mathtt{Base}$, as long as it is not set so large that the number of partitions is too small. Typically, we can set $\mathtt{Base} \in (1,5]$. For smaller datasets, a smaller $\mathtt{Base}$ is preferred to obtain more careful partitions.
>
> Table R5: Sensitivity analysis about the base of exponential partitions on CIFAR-10. ResNet-50 is trained for 200 epochs with three independent runs. Random$ ^ *$ denotes online random sampling. Test Accuracy (%) is reported.
>
> | Base         | 1.4                | 1.6                | 1.8                | 2                  | 5                  | 10                 | Random$ ^ {*}$     |
> | ------------ | ------------------ | ------------------ | ------------------ | ------------------ | ------------------ | ------------------ | ------------------ |
> | #Partitions  | 34                 | 25                 | 20                 | 17                 | 8                  | 6                  | 1                  |
> | Accuracy (%) | $92.7 _ {\pm 0.3}$ | $93.6 _ {\pm 0.3}$ | $93.0 _ {\pm 0.3}$ | $93.4 _ {\pm 0.3}$ | $92.8 _ {\pm 0.3}$ | $91.9 _ {\pm 0.3}$ | $85.6 _ {\pm 0.3}$ |

---

> ### Author Response · Authors · 2025-11-23
> **Response to Reviewer Hi6J (3/3)**
>
> **Q1 [Generalization to newer LLM settings]**
>
> **A.** Both the Alpaca dataset and Llama-7B that we used for fine-tuning were released in 2023. To further investigate the generalization of SRS-Sampling to newer LLM settings, we fine-tune *Mistral-7B* [1] (updated in July 2025) on the *LESS* dataset [2] (released in 2024), which is a combination of four instruction tuning datasets and comprises $270k$ data points. We set the subset ratio to **1%**, so as to examine the effectiveness of SRS-Sampling with an extremely small subset ratio, which is a potential application scenario, particularly in large-scale LLM fine-tuning. Benchmark results (Accuracy %) for SRS-Sampling and Random$ ^ *$ are reported below, all other fine-tuning settings follow [2].
>
> Table R6: Fine-tuning Mistral-7B on LESS for 15 epochs. Subset ratio is set to 1%. Random$ ^ *$ denotes online random sampling. Results for total node time for fine-tuning and zero-shot Accuracy (%) on four benchmarks are reported.
>
> | Method       | Time(h) | BBH      | HellaSwag | MMLU-Overall | MMLU-Hum. | MMLU-Other. | MMLU-Soc. | MMLU-STEM | Human-Eval | Avg.     |
> | ------------ | ------- | -------- | --------- | ------------ | --------- | ----------- | --------- | --------- | ---------- | -------- |
> | Random$ ^ *$ | 10.31   | 53.4     | 52.3      | 60.4         | 52.6      | 67.5        | 68.4      | 53.2      | 64.3       | 57.6     |
> | SRS-Sampling | 10.32   | **55.8** | **53.2**  | **61.7**     | **54.9**  | **68.6**    | **69.7**  | **53.7**  | **65.4**   | **59.0** |
>
> The results show that SRS-Sampling generalizes well to different LLM architectures and newer fine-tuning datasets. Moreover, in the scenarios with extremely small subset ratios, SRS-Sampling exhibits a more significant advantage over Random$ ^ *$ (vanilla uniform sampling), where the benchmark score (Accuracy) improvement is up to **1.4%p**.
>
> [1] Jiang, et al. “Mistral 7B.” *ArXiv* abs/2310.06825, 2023.
>
> [2] Xia, Mengzhou, et al. "LESS: selecting influential data for targeted instruction tuning." ICML, 2024.
>
>
>
> **Q4 [Performance on noisy datasets]**
>
> **A.** Thank you for this interesting question. We conduct additional experiments on CIFAR-10 with different ratios of label noise, where the subset ratio is set to 70% and we train ResNet-50 for 200 epochs. The accuracy (%) is reported over three runs. The results suggest that vanilla uniform sampling (Random$ ^ *$) is more vulnerable to noise compared to SRS-Sampling, which provides approximation error guarantees. SRS-Sampling enables the training of neural networks more robust to the label noise, and the advantage on test accuracy is more notable when the corruption ratio is higher.
>
> Table R7: Test Accuracy (%) of ResNet-50 with 200 epochs training from three independent runs on CIFAR-10 with label noise. Random$ ^ *$ denotes online random sampling.
>
> | Corruption Ratio | 0.1                | 0.2                | 0.3                | 0.4                | 0.5                |
> | ---------------- | ------------------ | ------------------ | ------------------ | ------------------ | ------------------ |
> | Random$ ^ *$     | $93.8 _ {\pm 0.3}$ | $88.6 _ {\pm 0.3}$ | $81.3 _ {\pm 0.3}$ | $77.0 _ {\pm 0.3}$ | $69.9 _ {\pm 0.3}$ |
> | SRS-Sampling     | $94.6 _ {\pm 0.3}$ | $89.5 _ {\pm 0.3}$ | $83.8 _ {\pm 0.3}$ | $80.3 _ {\pm 0.3}$ | $72.9 _ {\pm 0.3}$ |
>
>
>
> **Q5 [Training complexity during different training phases]** "Does the adaptive sample size based on stable rank sufficiently capture dynamic changes in training complexity during different training phases, e.g., early vs. late?"
>
> **A.** Thanks for this question. Let us first explain why we use stable rank, and then try to discuss the experimental observation.
>
> 1. **[Rationality for the stable rank analysis]** The main reason is that the required sample size depends heavily on the stable rank of the GT matrix, if we want to achieve the bounded approximation error over the entire **GT subspace**. In this paper, we establish the linear dependence between overall sample size and GT stable rank (Theorem 1).
> 2. **[Interpretation of experimental results]** In our paper, Figure 5 visualizes the stable rank ratio (stable rank compared to parameter dimension) of the gradient trajectory across training on CIFAR-10. The results show that **the stable rank upper bound across the early and late stages of training remains consistently low** compared to the parameter size. However, we did not observe an obvious monotonically increasing or decreasing trend in the experiment. In future, we think an interesting direction is to investigate how training complexity evolves across different training stages, through  a fine-grained stable rank analysis. In particular, we need to develop some new theoretical framework to characterize the evolving dynamics of stable rank throughout the training process.

---

> ### Author Response · Authors · 2025-11-27
>
> Thank you for taking the time to review our paper and provide valuable feedback. As the discussion phase nears its conclusion, we would like to confirm that our responses have addressed your concerns. If you have any additional comments, we will do our best to address them.

---

> > ### Comment · Reviewer_Hi6J · 2025-11-27
> >
> > Thank you for your response to the review. I am happy with the response and I have raised my score.

---

### Official Review · Reviewer_EiVZ · 2025-11-01

**Soundness:** 4
**Presentation:** 3
**Contribution:** 3
**Rating:** 8
**Confidence:** 3

**Summary:**

This paper introduces a novel and theoretically-grounded data selection method, named Stable Rank related Stratified Sampling (SRS-Sampling), for accelerating the training of deep neural networks. The key idea is that the effective dimensionality of model training is governed not by the parameter dimension but by the stable rank of the gradient trajectory matrix, which captures the intrinsic subspace of model updates. Building on this insight, this paper proposes a stratified sampling algorithm that allocates sampling ratios across loss strata in proportion to the estimated subspace complexity. Theoretical analysis shows that the required subset size scales linearly with the stable rank rather than with the parameter dimension. Experiments on CIFAR-10/100, ImageNet-1K, and LLaMA-7B fine-tuning demonstrate consistent acceleration and accuracy retention.

**Strengths:**

- **Strong theoretical insight**: The paper establishes a novel theoretical connection between the stable rank of the gradient trajectory and data sample complexity, extending beyond traditional parameter-dependent coreset theory.

- **Solid empirical validation**:
(1) Direct Theory Validation: The paper includes crucial experiments that directly support its theoretical claims. Figure 5, which visualizes the stable rank ratio during training, confirms the core premise that training dynamics are low-dimensional.
(2) Rich experiments demonstrated that SRS-Sampling show clear advantages over a wide array of strong baselines across multiple datasets and tasks.

**Weaknesses:**

(Major)

**Limited ablation on hyper-parameters**. The performance of SRS-Sampling may depend on the number of loss strata and the details of stable-rank estimation. A more extensive ablation would clarify the method’s sensitivity and robustness.

**Computational complexity of stable-rank estimation**.
Although the paper claims negligible runtime overhead (<0.1%), it does not formally analyze the computational complexity of stable rank estimation. A complexity expression or scalability discussion with respect to model dimension and training length would make the analysis more rigorous.

(Minor)

**Typographical and grammatical issues**. Several minor issues should be corrected for clarity and professionalism. For example: "gird" should be "grid" in the Figure 3 caption, and subject-verb agreement should be corrected (e.g., "...gradient structure has..." instead of "have" on Page 3, Line 158).

**Questions:**

- Could the authors provide an explicit complexity expression for stable rank estimation (e.g., O(k·d) with respect to iteration count k)?

- How sensitive is the method to the number of strata and the sampling ratio allocation strategy? Would adaptive stratum partitioning improve performance?

- Is the stable rank computed once before training, or does it evolve during training? If the latter, how often should it be updated in practice?

---

> ### Author Response · Authors · 2025-11-23
> **Response to Reviewer EiVZ (1/2)**
>
> We thank the reviewer for the constructive feedback. Below, we try to address your concerns.
>
> **W1 & Q2 [Ablation: sensitivity with respect to the number of loss strata and sampling ratio allocation strategy]** "How sensitive is the method to the number of strata and the sampling ratio allocation strategy? Would adaptive stratum partitioning improve performance?"
>
> **A.**
>
> 1. **[Exponential partitions]** The exponential partition is not a heuristic design; it is related to the bounded loss approximation error guarantee provided by SRS-Sampling. **First**, it directly enables the Hoeffding-based concentration bound for each loss stratum, as detailed in the proof of Lemma 2. The upper bound of loss value on each stratum is at most $\mathtt{Base}$ times of the lower bound, where $\mathtt{Base}=2$ by default in our experiments. It ensures that the sample size $|Q _ j|$ remains relatively small via Hoeffding's bound, as shown in Section 4.1 and in the complete proof of Theorem 1.  **Second**, the exponential partition ensures a bounded number of partitions $N = \lceil \log _ 2 n \rceil$, as shown in Line 205, where this logarithmic bound can be regarded as a relatively small number.
> 2. **[Sensitivity of number of loss strata]** The number of loss strata depends on the $\mathtt{Base}$ of the exponential partition.
>    Here we perform the sensitivity analysis with respect to the $\mathtt{Base}$ of exponential partitioning on the CIFAR-10 dataset using a ResNet-50 model, with three independent runs. The dataset size is $n=50000$, and the number of loss strata is $N+1=\lceil \log _ {\mathtt{Base}} n \rceil+1$. The subset ratio is $10\%$, and Random$ ^ {*}$ baseline is also included for comparison.
>    As validated by our experiments, the performance is relatively robust on the setting of $\mathtt{Base}$, as long as it is not set so large that the number of partitions is too small. Typically, we can set $\mathtt{Base} \in (1,5]$. For smaller datasets, a smaller $\mathtt{Base}$ is preferred to obtain more careful partitions.
>
> Table R4: Sensitivity analysis about the base of exponential partitions on CIFAR-10. ResNet-50 is trained for 200 epochs with three independent runs. Random$ ^ *$ denotes online random sampling. Test Accuracy (%) is reported.
>
> | Base         | 1.4                | 1.6                | 1.8                | 2                  | 5                  | 10                 | Random$ ^ {*}$     |
> | ------------ | ------------------ | ------------------ | ------------------ | ------------------ | ------------------ | ------------------ | ------------------ |
> | #Partitions  | 34                 | 25                 | 20                 | 17                 | 8                  | 6                  | 1                  |
> | Accuracy (%) | $92.7 _ {\pm 0.3}$ | $93.6 _ {\pm 0.3}$ | $93.0 _ {\pm 0.3}$ | $93.4 _ {\pm 0.3}$ | $92.8 _ {\pm 0.3}$ | $91.9 _ {\pm 0.3}$ | $85.6 _ {\pm 0.3}$ |
>
> 3. Finally, we would like to thank the reviewer for raising the "adaptive stratum partitioning" idea. The difficulty for realizing this idea mainly lies in how to provide the loss approximation guarantee while preserving relatively low sample size. We believe it is deserved to study in future and it may need to develop some new analyzing techniques.

---

> ### Author Response · Authors · 2025-11-23
> **Response to Reviewer EiVZ (2/2)**
>
> **W2 & Q1 [explicit complexity expression for stable rank estimation]**
>
> **A.** Thanks for this question. We do provide a complexity analysis in Section B.1 (Lines 1054-1079) and it is linked in the main text in Line 277. For stable rank upper bound estimation, Algorithm 2 runs in $\mathtt{Time} _ {sr}=O(d ^ {(l)}sk)$, which involves the Lanczos algorithm [1] for solving eigen-pairs. $d ^ {(l)}$ is the row size of the Kronecker product $S= C \otimes B$ ($\textit{i.e.,}$ the dimension of gradient vector of layer $l$), $s \leq d ^ {(l)}$ is the average number of nonzero elements in a row of $S$, and first $k\ll d ^ {(l)}$ eigen-pairs are computed (note that $1-\eta \lambda _ 1, 1-\eta \lambda _ 2$ are the two largest distinct eigenvalues of $I-\eta S$ as shown in the proof of Theorem 2. Therefore, we have $k=2$ iterations typically). By utilizing the basic property about eigen-pairs of Kronecker product (Theorem 4.2.12 of [2]), the problem size, i.e., $d ^ {(l)}$ of $S$ can be further decreased to the column size and row size of parameter matrix at layer $l$ (roughly speaking, the hidden size $h$ of $C$ and $B$). Therefore, the computational complexity can be further reduced to $O(k * h ^ 2)$. Typically we have $h ^ 2 = O(d/L)$ for an $L$ layers neural network with total parameter size of $d$. **Overall**, even if we disregard the parallel-processing capabilities of GPUs for matrix–vector multiplication, and sequentially compute for $L$ layers, the time complexity for Algorithm 2 is only $\mathtt{Time} _ {sr}=O(k * L*d/L)=O(kd)$. If we consider the parallel computation based on GPU, the computational complexity can be further reduced.
>
> [1] Calvetti, Daniela, Lothar Reichel, and Danny Chris Sorensen. "An implicitly restarted Lanczos method for large symmetric eigenvalue problems." *Electronic Transactions on Numerical Analysis*, 1994.
>
> [2] Roger A Horn and Charles R Johnson. "Topics in matrix analysis." Cambridge university press, 1994.
>
>
>
> **W3 [Typo issues]**
>
> **A.** Thank you for pointing out these typos. We will correct "gird" to “grid” in the Figure 3 caption, and "...gradient structure has..."  instead of "have" in Line 157.
>
>
>
> **Q3 [Practical frequency for stable rank updates]**
>
> **A.** The stable rank evolves during training. We update the weighted subset by Algorithm 1 at the beginning of each epoch. At the same time, the stable rank upper bound is computed by Algorithm 2 to determine the sample size in Algorithm 1 (once per epoch). The overall non-parallelized computational complexity $O(kd)$ of Algorithm 2 is relatively lightweight, which is practicable for real-time training; moreover, the $O(kd)$ complexity can be further reduced by fast GPU implementation. Therefore, for practical training, one can update the weighted subset together with the stable rank estimation **once per epoch**, where this setting is specified in Line 413.

---

> > ### Comment · Reviewer_EiVZ · 2025-11-26
> > **Rebuttal Comment**
> >
> > I thank the authors for their detailed response. The authors have addressed my prior concerns.
> >
> > Additionally, I find the 200-epoch ImageNet results (Table R1 from the rebuttal) very important. The significant performance gap there demonstrates the method's convergence efficiency under limited training budgets. I suggest adding these results to the paper.
> >
> > Finally, I am happy to maintain my positive rating.

---

> > > ### Author Response · Authors · 2025-11-27
> > >
> > > We will include Table R1 in our revised paper. Thank you sincerely for your helpful comments.

---

### Official Review · Reviewer_csr3 · 2025-11-04

**Soundness:** 2
**Presentation:** 2
**Contribution:** 2
**Rating:** 4
**Confidence:** 4

**Summary:**

- The authors propose to study the stable rank of the gradient trajectory matrix, which is a continuous and soft version of rank, newly defined in their paper as the rank of principal components in the gradient trajectory matrix.
- The authors suggest Stable Rank related Stratified Sampling for training data selection, for data-efficient training of deep learning models, where the suggested SRS sampling for training data selection depends on the initial parameter values.
- The authors conducted diverse experiments, including image classification with multiple datasets and LLM fine-tuning.

**Strengths:**

- The proposed SRS-sampling algorithm is straightforward and easy to understand.
- I haven't read all the proofs, but the provided proof sketches seem reasonable at least to me).

**Weaknesses:**

- Even the proposed SRS-sampling algorithm (Algorithm 1) is simple, but it is difficult to understand the completeness of the algorithm because it is hard to connect with Algorithm 2, which is linked within Algorithm 1.
- There should be error bars in the experimental results, since some performance gaps are tight.

**Questions:**

- Please explain Figure 1 in more detail. What are $\theta_T$ and $\theta_{2T}$? What are the green space and the orange space, respectively?
- Line 256: Why is the gradient norm in training a neural network typically bounded? Please specify the assumptions if they are required.
- Line 313: $G = A - BWC$ seems to have some meaning for each $A, B, C$. Could you explain this in more detail? (I know there is an extra explanation regarding this in the appendix as a previous work, but please provide me with an insight about this formulation.)
- In Theorem 2, $\lambda_1$ and $\lambda_2$ are the two smallest eigenvalues of $S$, right? Do they have to be non-negative or something? (And why?)
- typo in Line 352: upper-bound (6)
- Regarding Algorithm 2, so it seems that the upper bound of $sr(D_T)$ can be recursively integrated with each layer's upper bound of $sr(D_T^(l))$, right? I might have lost somewhere between, but why the "additionally" integrating the upper bound of $sr(D_T^(l))$?
- So the selected samples with the SRS perfectly depend on the initial neural network parameter values, where all other settings, including the network structure, loss function, etc., are given, right? Then, what proportion of the training dataset intersects (on average) for the different initialization?
- Could you compare the computational complexity for the baselines as well?

**Details Of Ethics Concerns:**

.

---

> ### Author Response · Authors · 2025-11-23
> **Response to Reviewer csr3 (1/5)**
>
> We thank the reviewer for the constructive feedback. Below, we try to address your concerns.
>
> **W1 [Connection between Algorithm 1 and Algorithm 2]**
>
> **A.** To clarify the relationship between Algorithm 1 and Algorithm 2, we emphasize two key aspects: the unbiased estimator and the bounded approximation error. Basically, Algorithm 1 is built upon the widely used importance sampling framework. However, importance sampling guarantees an unbiased estimate only for the loss evaluated at the current parameter $\theta _ {\mathtt{ini}}$. To provide a bounded approximation error for the objective function over the **whole** GT subspace, the difficulty lies in quantifying the required sample size $|Q _ j|$, whose value depends on $\mathrm{sr}(D _ T)$, as specified in Lemma 1. **Therefore**, we need Algorithm 2 to compute an upper bound on $\mathrm{sr}(D _ T)$, so as to determine the sample size in Algorithm 1 (note that the required sample size increases with $\mathrm{sr}(D _ T)$, as shown by Lemma 1 and Theorem 1).

---

> ### Author Response · Authors · 2025-11-23
> **Response to Reviewer csr3 (2/5)**
>
> **W2 [Error analysis and performance gaps in experiments]**
>
> **A.** Thank you for the suggestion.
>
> 1. To address your concern, we report the detailed standard deviation error analysis together with **p-values from paired t-tests** across three independent runs for 200 epochs, where other setting is the same as the ImageNet-1K experiment in Table 3.
>
> **Performance Results (Mean ± Std)**
>
> Table R1: ImageNet-1K results of standard deviation error analysis from three independent runs. Three models are trained from scratch for 200 epochs, ViT-B is additionally fine-tuned for 50 epochs. Random$ ^ *$ denotes online random sampling.
>
> | Model     | Subset Ratio | Random*       | InfoBatch     | SRS-Sampling      |
> | --------- | ------------ | ------------- | ------------- | ----------------- |
> | ResNet-50 | 60.0%        | 72.48 ± 0.438 | 74.86 ± 0.437 | **76.56 ± 0.252** |
> | Swin-T    | 80.0%        | 75.71 ± 0.435 | 78.14 ± 0.385 | **80.84 ± 0.311** |
> | ViT-B     | 80.0%        | 79.04 ± 0.454 | 80.78 ± 0.420 | **82.36 ± 0.245** |
>
>
>
> **Statistical Significance (p-values)**
>
> Table R2: ImageNet-1K results of p-values from paired t-tests from three independent runs. Three models are trained from scratch for 200 epochs, ViT-B is additionally fine-tuned for 50 epochs. Random$ ^ *$ denotes online random sampling.
>
> | Model     | SRS-Sampling vs Random* | SRS-Sampling vs InfoBatch |
> | --------- | ----------------------- | ------------------------- |
> | ResNet-50 | 0.0093 (<0.01)          | 0.0482 (<0.05)            |
> | Swin-T    | 0.0059 (<0.01)          | 0.0004 (<0.001)           |
> | ViT-B     | 0.0128 (<0.05)          | 0.0273 (<0.05)            |
>
> Our paired t-test analysis suggests that SRS-Sampling achieves statistically significant improvements over both Random$ ^ {*}$ and InfoBatch baselines across all three models (ResNet-50, Swin-T, and ViT-B) in large-scale ImageNet-1K training, with all p-values below $0.05$, which validates the statistical significance.
>
> 2. For the `Full Data` baseline in Table 2, the tight performance gap relative to it highlights the effectiveness of our method. Specifically, Table 2 shows that the proposed SRS-Sampling method achieves **nearly lossless performance** while using only **60%** of the data on the **large-scale dataset (ImageNet-1K)**. Meanwhile, the results with **standard deviations** such as "$\pm 0.2$"  are also reported in **Table 2**. Additional standard deviation analysis can be found in the appendix, please see **Table 9 and Table 10** in Section C.6. **Table 8** shows the results on CIFAR-10 and CIFAR-100, which suggests that SRS-Sampling provides a significant advantage over Random$ ^ *$ in the early stage of training, achieving up to **12.5%p improvement** at 40% of the training process. This indicates the effectiveness of SRS-Sampling in accelerating convergence, particularly when early-stage gradients may be too chaotic for vanilla uniform sampling.
> 3. To further validate the advantage of SRS-Sampling, we fine-tune *Mistral-7B* [1] (updated in July, 2025) on *LESS* dataset [2] (released in 2024), which is a combination of four instruction tuning datasets and contains $270k$ data points. We set the subset ratio to be 1%, so as to examine the effectiveness of SRS-Sampling with an extremely small subset ratio, which is a potential application scenario, particularly in **large-scale LLM fine-tuning**.
>
> Table R3: Fine-tuning Mistral-7B on the LESS dataset for 15 epochs. Subset ratio is set to 1%. Random$ ^ *$ denotes online random sampling. Results of total node time for fine-tuning and Zero-shot Accuracy (%) on four benchmarks are reported.
>
> | Method       | Time(h) | BBH      | HellaSwag | MMLU-Overall | MMLU-Hum. | MMLU-Other. | MMLU-Soc. | MMLU-STEM | Human-Eval | Avg.     |
> | ------------ | ------- | -------- | --------- | ------------ | --------- | ----------- | --------- | --------- | ---------- | -------- |
> | Random$ ^ *$ | 10.31   | 53.4     | 52.3      | 60.4         | 52.6      | 67.5        | 68.4      | 53.2      | 64.3       | 57.6     |
> | SRS-Sampling | 10.32   | **55.8** | **53.2**  | **61.7**     | **54.9**  | **68.6**    | **69.7**  | **53.7**  | **65.4**   | **59.0** |
>
> The results show that SRS-Sampling generalizes well to different LLM architectures and newer fine-tuning datasets. Moreover, in the scenarios with extremely small subset ratios, SRS-Sampling exhibits more significant advantage over Random$ ^ *$ (vanilla uniform sampling), where the benchmark score improvement is up to **1.4%p**.
>
> [1] Jiang, et al. “Mistral 7B.” *ArXiv* abs/2310.06825, 2023.
>
> [2] Xia, Mengzhou, et al. "LESS: selecting influential data for targeted instruction tuning." ICML, 2024.

---

> ### Author Response · Authors · 2025-11-23
> **Response to Reviewer csr3 (3/5)**
>
> **Q1 [Detailed explanation for Figure 1]**
>
> **A.** Figure 1 is an illustration of how the local gradient trajectory (GT) mainly resides in a low-dimensional principal space. Specifically, the notation $\theta _ t$ indicates the parameter in iteration $t$, and $\theta _ 0$ is the initialization. Every $T$ iterations we construct the weighted subset by Algorithm 1 (e.g., at parameters $\theta _ {0}, \theta _ {T}, \theta _ {2T}$ as "$\theta _ \mathtt{ini}$" in Algorithm 1), where $T$ is set to be the number of iterations in one epoch in our experiments, with this setting specified in Line 413. The green and orange spaces are the principal spaces spanned by the top-$r$ singular vectors of the GT matrices (as mentioned in Line 119). The dimension $r$ of the principal space is characterized by the stable rank of the GT matrix (Section 4.1), whose upper bound is computed via Algorithm 2 using the corresponding $\theta _ \mathtt{ini}$.
>
>
>
> **Q2 [Assumption about gradient norm]**
>
> **A.** We do specify the assumption of maximum gradient norm $M$ in Line 261 and incorporate it into our analysis in Section 4 (Theorem 1). It is usually reasonable to assume the existence of a bounded $M$, since unbounded gradient norm can lead to the exploding gradient problem and training failures. One can ensure bounded gradient norms by using techniques like the popular *gradient Clipping* [1] in practice.
>
>
> [1] Pascanu, Razvan, Tomas Mikolov, and Yoshua Bengio. "On the difficulty of training recurrent neural networks." ICML, 2013.
>
>
>
> **Q3 [A more detailed discussion on the gradient structure]**
>
> **A.** Intuitively, the gradient structure $G=A-BWC$ measures the gap between the label-related component $A$ and the parameter-related component $BWC$ in the gradient space, where $B$ and $C$ can be seen as the left and right projection matrices of mapping from the parameter space to the gradient space. Specifically, for a layer parameterized by matrix $W$, the input-dependent matrix $A$ captures the label signal $y$ separately, since $y$ only exists in $A$. The left projection matrix $B$ captures the backward flow of the neural network, since $B$ only involves the Jacobian $J _ l$ after this layer. And the right projection matrix $C$ captures the forward flow of the neural network, since $C$ only involves the activation $f _ {l-1}$ before this layer.

---

> ### Author Response · Authors · 2025-11-23
> **Response to Reviewer csr3 (4/5)**
>
> **Q4 [Eigenvalues of $S$]**
>
> **A.** $\lambda _ 1$ and $\lambda _ 2$ are the two smallest non-zero and distinct eigenvalues of $S$. **First**, we show that **all eigenvalues of S are non-negative**. From the derivation of gradient structure in Theorem 3 and Theorem 4, we can conclude that $B _ i,C _ i$ are Positive Semi-Definite (PSD) matrices (as mentioned in Line 340), consequently their Kronecker product $C _ i \otimes B _ i$ is PSD. Since the weight $\mu _ i$ is non-negative from our construction in Algorithm 1, $S$ is also PSD, and all eigenvalues are non-negative. **Second**, we are particularly interested in finding the two smallest, non-zero and distinct $\lambda _ 1, \lambda _ 2$. The reason for specifying "non-zero, distinct" is that we need to distinguish the eigen-spaces with different convergent rates (as detailed in Lines 350-355) such that a meaningful bound on the stable rank can be provided in Theorem 2.
>
>
>
> **Q5 [Typo fix]**
>
> **A.** Thank you. We will correct “upper-bound 6” to “upper-bound (6)” in the revision.
>
>
>
> **Q6 [Explanation for the summation of $\mathrm{sr}(D _ T ^ {(l)})$ in  Algorithm 2]**
>
> **A.** The sample size specified in Theorem 1 depends on $\mathrm{sr}(D _ T)$ (where $D_T$ comes from the recursive expression for $\{ g _ t \} _ {t=0}^{T-1}$ as detailed in the proof of Theorem 2). The value of $\mathrm{sr}(D_T)$ reflects the dimension of the principal subspace of the gradient trajectory, where the gradient is taken with respect to the whole neural network instead of a single layer. Notice that the gradient spaces across different layers are disjoint (i.e., different "slices" of the gradient vector), and consequently their respective principal subspaces are also disjoint. Therefore, $\mathrm{sr}(D _ T)$ can be computed as the sum of the stable ranks for each layer, i.e., $\mathrm{sr}(D_T)=\sum_{l=1}^{L} \mathrm{sr}(D _ T ^ {(l)}) $ as in Algorithm 2.
>
>
>
> **Q7 [Proportion of subset intersection]**
>
> **A.**
>
> Thanks for this question. Actually, when running Algorithm 1 with *different* $\theta _ {\mathtt{ini}}$, the proportion of subset intersection can be arbitrary. This is because that the same data item $(x _ i,y _ i)$ can have different loss values under varying $\theta _ {\mathtt{ini}}$, and it might belong to a different $P _ j$. This changes $|P_j|$ as well as the probability that $(x_i, y_i)$ is sampled.  But we would like to emphasize that the proportion of the subset intersection is not an issue in our method. Our major theoretical result Theorem 1  provides the theoretical guarantee that the objective function can be faithfully approximated using the selected weighted subset, and the proof is independent of the subset intersection.

---

> ### Author Response · Authors · 2025-11-23
> **Response to Reviewer csr3 (5/5)**
>
> **Q8 [Complexity comparison with baselines]**
>
> **A.**
>
> 1. **Our complexity** The computational complexity of SRS-Sampling is $O(n+kd)$, where $n$ is the size of the dataset, $k$ is the number of eigen-pairs to solve, and $d$ is the number of parameters in a neural network.
>
>    Specifically, for stable rank upper bound estimation, Algorithm 2 runs in $\mathtt{Time} _ {sr}=O(d ^ {(l)}sk)$, which involves the Lanczos algorithm [1] for solving eigen-pairs. $d ^ {(l)}$ is the row size of the Kronecker product $S= C \otimes B$ ($\textit{i.e.,}$ the dimension of gradient vector of layer $l$), $s \leq d ^ {(l)}$ is the average number of nonzero elements in a row of $S$, and first $k\ll d ^ {(l)}$ eigen-pairs are computed (note that $1-\eta \lambda _ 1, 1-\eta \lambda _ 2$ are the two largest distinct eigenvalues of $I-\eta S$ as shown in the proof of Theorem 2. Therefore, we have $k=2$ iterations typically). By utilizing the basic property about eigen-pairs of Kronecker product (Theorem 4.2.12 of [2]), the problem size, i.e., $d ^ {(l)}$ of $S$ can be further decreased to the column size and row size of parameter matrix at layer $l$ (roughly speaking, the hidden size $h$ of $C$ and $B$). Therefore, the computational complexity can be further reduced to $O(k * h ^ 2)$. Typically we have $h ^ 2 = O(d/L)$ for an $L$ layers neural network with total parameter size of $d$. **Overall**, even if we disregard the parallel-processing capabilities of GPUs for matrix–vector multiplication, and sequentially compute for $L$ layers, the time complexity for Algorithm 2 is only $\mathtt{Time} _ {sr}=O(k * L*d/L)=O(kd)$. If we consider the parallel computation based on GPU, the computational complexity can be further reduced.
>
> 2. **Baselines** Among the baselines, InfoBatch has a computational complexity of $O(n)$.  K-Center, Least Confidence, Margin, CD, Craig, GREATS, Glister take $O(bnd)$ running time, for the selection of $b$ samples from a dataset of size $n$. Influence method takes $O(nd \kappa)$ running time, where $\kappa$ is the factor for inverse Hessian approximation. DP takes $O(nd \kappa + T _ {\mathtt{iter}})$ for additionally solving a constraint problem. UCB, $\epsilon-$greedy, EL2N, Forgetting and GraNd take $O(nd + b\log n)$ running time, including the cost of computing data scores and selecting the top-$b$ samples.
>    **Overall**, apart from Random$^*$ which just performs simple uniform sampling, SRS-Sampling's $O(n+kd)$ complexity is only slightly higher than InfoBatch's $O(n)$ complexity (as mentioned above, we typically take $k=2$ for solving two distinct eigen-pairs, and the extra $O(kd)$ complexity can be further reduced by fast GPU implementation).
>
> [1] Calvetti, Daniela, Lothar Reichel, and Danny Chris Sorensen. "An implicitly restarted Lanczos method for large symmetric eigenvalue problems." *Electronic Transactions on Numerical Analysis*, 1994.
>
> [2] Roger A Horn and Charles R Johnson. "Topics in matrix analysis." Cambridge university press, 1994.

---

> ### Author Response · Authors · 2025-11-27
>
> Thank you for taking the time to review our paper and provide valuable feedback. As the discussion phase nears its conclusion, we would like to confirm that our responses have addressed your concerns. If you have any additional comments, we will do our best to address them.

---

### Author Response · Authors · 2025-11-23
**General Response**

We sincerely thank all the anonymous reviewers for their valuable comments. We have included additional content to our revised paper based on the comments and questions. The changes are summarized below and marked in **blue** within the paper. **The line numbers refer to those in our revised PDF**.

Revisions to the paper

* Page 2, Figure 1 caption. We have updated a detailed explanation for Figure 1, especially for the principal subspace (for **Q1** from reviewer `csr3`).
* Page 6, Line 274-278; Page 20, Lines 1054-1079. We provide an explicit complexity expression for stable rank estimation (for **Q1 & W2** from reviewer `EiVZ`, **W2 & Q3** from reviewer `Hi6J`, and **Q3** from reviewer `6krn`).
* Page 6, Lines 317-320. We provide a more detailed discussion on the gradient structure for better understanding (for **Q3** from reviewer `csr3`).
* Page 9, Lines 449-452, Table 5; Page 27, Lines 1445-1450; Page 28, Lines 1480-1483; Page 30, Lines 1578-1582. We add an experiment that fine-tunes Mistral-7B on the LESS dataset with 1% subset ratio (for **Q1** from reviewer `Hi6J`, **Q1** from reviewer `6krn`).
* Page 9, Lines 485-500, Table 6. We add the sensitivity analysis for the base of exponential partitioning in SRS-Sampling (for **W1 & Q2** from reviewer `EiVZ`, **W3 & Q2** from reviewer `Hi6J`, **Q2** from reviewer `6krn`).
* Page 30, Lines 1605-1619 Table 11. We add an experiment for noisy datasets (for **Q4** from reviewer `Hi6J`).

---

### Author Response · Authors · 2025-12-03
**Final summary to PCs, SACs, ACs, and Reviewers**

Dear PCs, SACs, ACs, and Reviewers,

We sincerely appreciate your diligent efforts and insightful feedback on our submission. We carefully considered each concern and responded to them in detail.

* Reviewer `Hi6J` was happy to raise the score **from 4 to 6** , as the concerns have been addressed.
* Reviewer `EiVZ` was happy to **maintain the positive rating of 8**, highlighting the **convergence efficiency** under limited training budgets.
* No further concern was raised by Reviewers `csr3` and `6krn`.

Below, we summarize the main concerns and our responses; they are also **affirmed as solved** by the reviewers engaged in discussion.


---
**[Common Concerns Across Reviewers]**

| **Common Concerns**                                          | **Raised By**                                                | **Our Consolidated Responses**                               | **Response Affirmed By** |
| ------------------------------------------------------------ | ------------------------------------------------------------ | ------------------------------------------------------------ | ------------------------ |
| Connection between GT analysis, Algorithms 1 & 2, and exponential partitions. | `csr3` (**W1**), `Hi6J` (**W1**).                            | We theoretically link GT stable rank by Algorithm 2 to per-stratum sample sizes in the exponential partitions of Algorithm 1. | `Hi6J`                   |
| Statistical significance and convergence advantages.         | `csr3` (**W2**), `6krn` (**W1**).                            | Added the experiments with error bars and $p$-values, showing that SRS-Sampling consistently exhibits statistically significant gains, faster convergence, and near full-data accuracy. | `EiVZ`                   |
| Computational and storage complexity.                        | `csr3` (**Q8**), `EiVZ` (**W2** & **Q1**, **Q3**), `Hi6J` (**W2** & **Q3**), `6krn` (**Q3**). | We demonstrate that stable-rank estimation adds only $O(kd)$ time (typically $k=2$) and $O(d)$ memory, making it efficient for real-time training. | `Hi6J`, `EiVZ`           |
| Design and sensitivity of exponential partitions.            | `EiVZ` (**W1** & **Q2**), `Hi6J` (**W3** & **Q2**), `6krn` (**W2** & **Q2**). | We justify the exponential partitioning via theoretical benefit of relatively small sample size and confirm its robustness via sensitivity studies. | `Hi6J`, `EiVZ`           |
| Generalization to challenging regimes (limited training budgets, extremely small subsets, new LLMs, noisy datasets). | `csr3` (**W2**), `Hi6J` (**Q1**, **Q4**), `6krn` (**Q1**).   | The results on 200-epoch ImageNet, Mistral-7B on $1\%$ LESS dataset, and noisy CIFAR-10 show the advantages of SRS-Sampling under those challenging regimes. | `Hi6J`, `EiVZ`           |


---
**[Strengths emphasized by Reviewers]**

| Category                     | Strengths                                                    | Reviewers              |
| ---------------------------- | ------------------------------------------------------------ | ---------------------- |
| **Theoretical Contribution** | Novel theoretical connection between stable rank and data sample complexity | `EiVZ`                 |
|                              | Novel idea of tracking dimensionality of gradient trajectory subspace for data selection, unaffected by parameter dimension | `6krn`, `Hi6J`         |
|                              | Clear theoretical foundation with reasonable proof sketches  | `csr3`, `6krn`         |
| **Empirical Validation**     | Direct theory validation confirming low-dimensional training dynamics | `EiVZ`                 |
|                              | Rich and diverse experiments across multiple datasets and tasks | `csr3`, `EiVZ`         |
|                              | Empirical results show the value and convergence efficiency  | `Hi6J`, `EiVZ`         |
| **Method Practicality**      | Straightforward and easy-to-understand algorithm             | `csr3`                 |
|                              | Smart stratified sampling approach, ensuring training speedups without losing accuracy | `Hi6J`                 |
|                              | Practical for large neural networks                          | `Hi6J`                 |
|                              | Efficient computation methods                                | `6krn`                 |
| **Presentation & Clarity**   | Well-written manuscript with clear and reasonable theoretical evidences | `csr3`, `6krn`         |
|                              | Clear motivation by outlining the limitations of prior works and challenges, easy to follow | `6krn`                 |
|                              | Good presentation                                            | `Hi6J`, `EiVZ`, `6krn` |



We hope these summary tables are helpful to evaluate our manuscript.

Sincerely,

The authors

---

### Meta-Review · Area_Chair_ebVq · 2026-01-06

**Summary:**

The submission studies online data selection for faster training by repeatedly forming a weighted subset that aims to approximate the full-data objective. The proposed approach partitions examples into exponentially spaced loss bins, computed at the start of each epoch, and then performs stratified sampling with importance weights, refreshing the subset once per epoch. The main conceptual claim is that the required subset size should scale with the effective dimensionality of training dynamics, measured by the stable rank of a gradient-trajectory matrix, rather than with the raw parameter dimension. The paper provides a routine to estimate an upper bound and use it to set sampling budgets.

I recommend rejection. The direction is promising, and the paper is thoughtful, but the current evidence and positioning do not yet support the strongest framing. The method remains closely aligned with existing loss-based weighted sampling schemes, and the stable-rank component currently reads more like a principled budget heuristic than a demonstrably necessary ingredient. The experimental results are encouraging, but they do not yet make a clear case that this is reliably more than a well-tuned variant across a broad range of realistic training regimes.

**Reviewer Concerns:**

The central reviewer concerns fall into two themes. One is practical efficiency and overall cost-benefit in realistic regimes. The paper argues that stable-rank estimation adds only a small overhead and can reuse values already computed during training, and the reported overhead appears small in the presented settings. The authors further clarified the complexity in the rebuttal.

The second theme is whether the approach is genuinely principled or mainly a tuned heuristic. Several reviewers found the connection between the gradient-trajectory analysis and the particular loss-based exponential stratification difficult to follow. The rebuttal clarifies the intended linkage and provides sensitivity results for the partition base; however, the overall design still resembles a standard importance-sampling template augmented with a stable-rank-driven budget rule rather than a sampling rule that naturally follows from the stable-rank analysis. This raises a reasonable doubt about the essentialness of stable-rank tracking to the observed gains and whether similar outcomes could be achieved by simpler loss-based scheduling rules. A stronger resolution would require clearer positioning against closely related online selection methods and a cleaner set of ablations that isolate the stable-rank driven decisions across multiple demanding regimes.

**Reviewer Scores:**

Hi6J initially scored 4 and later indicated satisfaction, increasing their score to (possibly) 6. With reviewer EiVZ keeping their score to 8, this ties the reviewer vote, and the decision then hinges on whether csr3 and 6krn are persuaded on the core contribution. My expectation is that both would likely remain at 4, because while the rebuttal strengthens confidence that the reported gains are statistically meaningful, it does not fully resolve their deeper concern that the method is very close to prior loss-based weighted sampling and that the stable-rank component is not yet shown to be essential rather than a principled tuning rule.

---

### Decision · Program_Chairs · 2026-01-26

Reject